# REFORMULATION FOR PRETRAINING DATA AUGMENTATION

**Xintong Hao[1], Ruijie Zhu[2]\*, Ge Zhang[1], Ke Shen[1]†, Chenggang Li[1]**
[1]ByteDance Seed   [2]University of California, Santa Cruz
{haoxintong, shenke}@bytedance.com

## ABSTRACT

Despite the impressive capabilities of large language models across various tasks, their continued scaling is severely hampered not only by data scarcity but also by the performance degradation associated with excessive data repetition during training. To overcome this critical bottleneck, we introduce the Massive Genre-Audience (**MGA**) reformulation method, a framework designed to augment corpora in a way that supports more effective model performance scaling. Instead of relying on complex, predefined seed systems, MGA systematically reformulates existing corpora into diverse, contextually-rich variations by adaptively generating genre-audience pairs. We present this framework and the resulting 770 billion token MGACorpus[1], created as a practical instantiation of our methodology. We experimentally validate MGA's core benefits by demonstrating superior scaling properties, in terms of both model size and data budget, against data repetition and upsampling (up to 13B parameters). Furthermore, our comprehensive analysis investigates the role of synthesis principles in generation quality and reveals nuances in evaluating model capabilities using standard loss metrics. Our work shows that a systematic framework like **MGA** provides a reliable pathway to substantially augment training datasets, effectively alleviating repetition bottlenecks and enabling more efficient scaling of large language models.

## 1 INTRODUCTION

The remarkable success of Large Language Models (LLMs) heavily relies on the scale of model parameters and training data (Kaplan et al., 2020; Hoffmann et al., 2022). Scaling laws demonstrate that improvements in model performance are increasingly dependent on data quantity and quality. However, the growth rate of available natural language corpora significantly lags behind the increasing demand for training data (Villalobos et al., 2022). While data repetition is a standard tool in traditional deep learning, it backfires in LLM pre-training, where it degrades performance and creates a major scaling bottleneck. This raises a critical question: how can we fully utilize the potential of existing data in data-constrained situations?

Leveraging LLMs to synthesize high-quality training data has emerged as a frontier approach (Su et al., 2024; Abdin et al., 2024). In theory, data synthesis can generate limitless training material, expanding datasets without the negative consequences of repetition. However, while the promise of synthetic data is clear, the specific methodologies, the **'how'** behind successful large-scale data synthesis often remain opaque, existing as black-box processes within large industrial labs rather than as systematic, reproducible science. Many prevailing methods depend on large-scale models for generation, effectively creating "distillations" rather than true data augmentations, or require sophisticated, pre-defined seed curation systems (Abdin et al., 2024; Ben Allal et al., 2024). These dependencies introduce substantial computational bottlenecks and limit their accessibility and scalability for the broader research community.

In this work, we propose **MGA** (Massive Genre-Audience reformulation), a transparent and principled framework designed to directly address the data repetition challenge by augmenting the raw

---

\*Work done at ByteDance Seed
†Corresponding author
[1]https://huggingface.co/datasets/ByteDance-Seed/mga-fineweb-edu

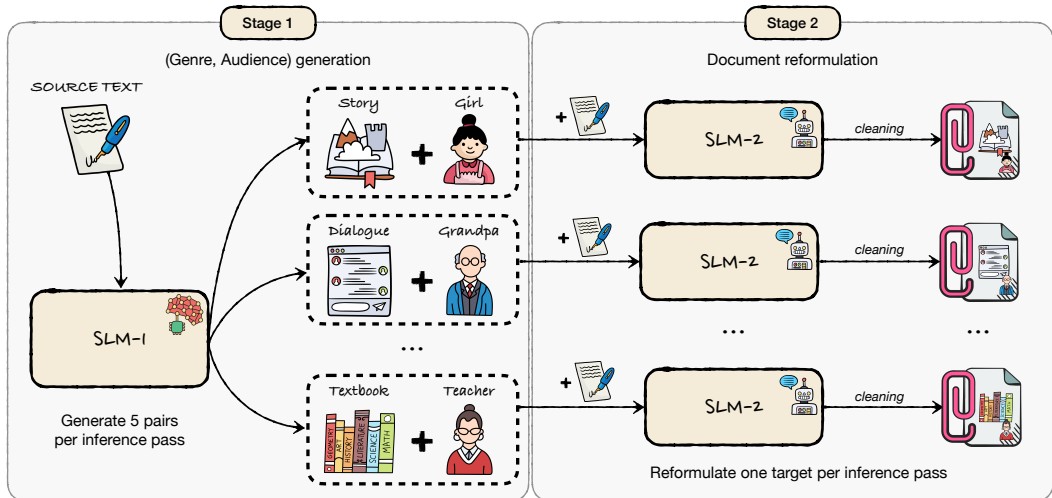

Figure 1: Overview of the **MGA** framework. Our framework expands the original corpus through a two-stage synthesis followed by a cleaning stage process. Each document is reformulated to 5 new documents, finally achieving a 3.9× token number expansion while maintaining diversity through adaptively generated (genre, audience) pairs.

text and creating more unique tokens. As illustrated in Figure 1, the **MGA** framework is efficiently implemented using a lightweight 3.3B MoE model. Crucially, it avoids complex external seed systems by adaptively generating diverse genre-audience pairs directly from raw input documents. This design makes the data generation process highly efficient and applicable to web-scale corpora.

However, proposing a framework is only the first step. To establish its scientific merit and provide actionable insights for the community, a deeper investigation is essential. We contend that a thorough understanding requires answering three fundamental questions: 1) How does MGA reformulation complement existing synthetic data strategies? 2) What is the core mechanism, specifically the role of diversity, that drives its effectiveness in data-scarce scenarios (especially high repetition scenarios)? 3) Why does reformulation fundamentally benefit the model's learning process? By addressing these questions, our main contributions are:

- We introduce the **MGA framework**, a systematic and reproducible methodology for corpus reformulation. To validate our framework and ensure full reproducibility, we will release the **MGACorpus** (a 770B token dataset) and open-source all key artifacts, including prompts, tool-model finetuning data, and cleaning scripts. Our experiments demonstrate that models trained on MGACorpus significantly outperform those trained on the original corpus it expands upon.

- We experimentally validate MGA's superior scaling properties in terms of both model size and data budget, revealing a widening performance gap over standard data repetition and upsampling across a wide range of model sizes (377M/1.7B/7B/13B) and data budget (up to 700B tokens).

- We analyze synthetic data collapse from two key perspectives: first, we characterize how synthesis principles (manifested through prompt engineering) mitigate collapse, and second, we reveal the limitations of validation loss as a collapse detection metric. This analysis provides key insights for future synthetic data optimization.

## 2 RELATED WORK

**Data Curation**  While web-crawled data contains hundreds of trillions of tokens, stringent quality filters typically remove the majority of this content. Popular datasets like C4, Gopher, Dolma, and RefinedWeb (Raffel et al., 2020; Rae et al., 2021; Penedo et al., 2023; Soldaini et al., 2024) use non-learned heuristics. More recently, aggressive model-based and retrieval-based filtering has become prominent in datasets like FineWeb-Edu (Penedo et al., 2024), DCLM (Li et al., 2024), and FineFineWeb (Zhang et al., 2024). Such heavy filtering results in a removal of over 90% of tokens,

which has led some researchers to focus on balancing accuracy and data quantity (Su et al., 2024). However, this does not alter the fact that the total amount of high-quality data remains limited.

**Repetition Training**  Studies on subset repetition training have revealed that model divergence tends to occur earlier as model parameters increase (Hernandez et al., 2022). For scenarios where entire datasets are repeated for training, limiting to 4 epochs or fewer results in minimal efficiency degradation (Muennighoff et al., 2023; Taylor et al., 2022). Researchers have explored regularization techniques to mitigate repetition degradation, but this highlights the critical need for careful hyperparameter tuning. For example, while some work shows that increasing weight decay can yield better metrics (Fang et al., 2025), the same technique can also destabilize training. In a set of ablation studies, Xue et al. (2023) found that models with weight decay failed to converge, whereas using dropout proved to be an effective alternative. This sensitivity underscores the challenge of applying regularization in repetition scenarios. Overall, this topic remains understudied across different hyper-parameters, data distributions, and repetition ratios.

**Synthetic Pretraining Data**  Data synthesis for pretraining has rapidly evolved, with two primary approaches: seed-based synthesis and raw-text rephrasing. Seed-based methods, exemplified by Phi models (Abdin et al., 2024), Cosmopedia (Ben Allal et al., 2024), use predefined seed systems and templates to precisely control the generated content. In contrast, rephrasing methods, such as WRAP (Maini et al., 2024) and Nemotron-CC (Su et al., 2024), rewrite existing web content into different formats. Recent state-of-the-art models have validated the effectiveness of rewriting at an unprecedented scale. The Kimi K2 model applied rewriting to knowledge data as part of its training corpus (Kimi et al., 2025), and Nemotron Nano 2 (NVIDIA et al., 2025) employs Qwen3-30B-A3B (Yang et al., 2025) for rewriting. Concurrently, recent analysis has begun to codify the high-level insights for success and highlight the importance of the 'methodology' behind data synthesis (DatologyAI et al., 2025). These efforts confirm that synthetic data is a key component in training frontier models.

While these works validate the general approach, they often do not provide the detailed information and ablation of a successful synthesis implementation. Our work bridges this gap by providing a concrete instantiation of these principles. Inspired by Maini et al. (2024); Ge et al. (2024), our framework **adaptively** generates diverse 'Genre' and 'Audience' pairs from high-quality source text. This approach systematically **enhances diversity in a scalable manner**, without requiring complex external seed systems or large-scale generator models.

## 3  MASSIVE GENRE-AUDIENCE REFORMULATION

The central challenge of data reformulation is balancing two competing goals: generating novel, diverse content (**variance**) while preserving the source document's core factual information (**invariance**). To resolve this tension, we introduce the Massive Genre-Audience (MGA) framework, a principled pipeline designed for systematic corpus expansion. The framework operates on our central principle of **"Limited Consistency"**, which seeks to balance stylistic diversity with factual fidelity. This approach is implemented efficiently using lightweight small language models (Tool SLMs) fine-tuned for specific sub-tasks, ensuring both quality and scalability.

### 3.1  LIMITED CONSISTENCY

We define "Limited Consistency" as a guiding principle that seeks to maximize the stylistic and structural variance of reformulated content while maintaining strict invariance of the source document's core factual information. This principle directly addresses the risk of generating data that is either too repetitive or factually incorrect. The primary mechanism for implementing this principle is through careful Prompt Engineering, which steers the generative process.

To identify an optimal balance, we explored the design space of PE strategies. Prompts that are too strict ('SLM-Strict') enforce high fidelity, leading to a distribution that closely mirrors the original corpus but lacks diversity. Conversely, prompts that are too relaxed ('SLM-Relaxed') encourage excessive deviation, resulting in a significant distributional shift and a high risk of factual degradation. Our final approach ('SLM-Base') is calibrated to strike a balance, expanding the original data distribution without losing topical coherence. The distinct distributional impacts of these strategies

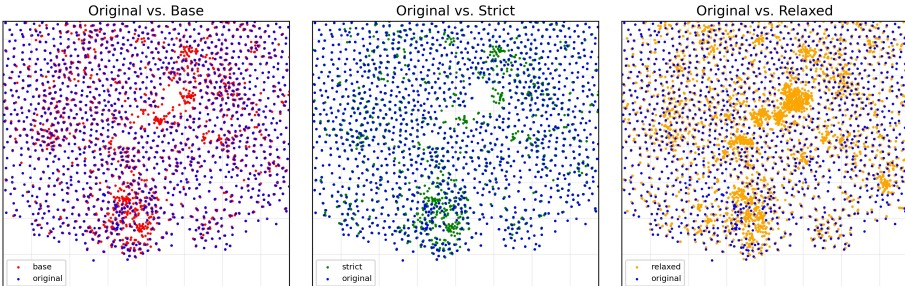

Figure 2: t-SNE visualization illustrating our "Limited Consistency" principle. Our Base PE strategy (left) achieves a balanced expansion of the original data distribution, while the Strict (middle) and Relaxed (right) variants are overly conservative or deviate excessively, respectively.

are visualized in Figure 2. A detailed quantitative analysis validating the superiority of our balanced approach is presented in the ablation studies in Section 4.3.2.

## 3.2 FRAMEWORK IMPLEMENTATION

The MGA framework is operationalized as a two-stage synthesis pipeline: a variance-maximizing stage for generating diverse directives, followed by an invariance-enforcing stage for controlled reformulation. Each stage is powered by a specialized Tool SLM, which is finetuned on task-specific data generated by a larger language model[2]. This implementation choice is validated in Table 1, where our final Tool SLM achieves performance nearly identical to the original LLM labeler.

Table 1: Reformulation quality comparison between the Tool SLM and its LLM teacher. All outputs were scored on a 1-5 scale by the LLM itself to evaluate the SLM's alignment. Scoring reliability was verified via human-in-the-loop cross-checking, yielding an alignment rate of over 90%.

| Models | Total Examples | 5 | 4 | 3 | 2 | 1 | Rate($\geq 3$) | Diff |
|---|---|---|---|---|---|---|---|---|
| Labeler LLM | 15,355 | 4,120 | 7,143 | 3,034 | 661 | 214 | 93.11% | - |
| Tool SLM | 15,355 | 3,788 | 7,124 | 3,224 | 736 | 285 | 92.06% | -1.05% |

**Stage 1: Adaptive GA-Pair Generation.** The primary objective of this stage is to **maximize diversity** by generating a wide array of creative reformulation directives. Our choice of Genre-Audience (GA) pairs as the core mechanism is deliberate. While simple rephrasing can generate stylistic variants, it often lacks structured diversity. GA pairs provide a robust framework for meaningful content adaptation:

- **Genre** dictates the structural and stylistic format of the content (e.g., an analytical report, a step-by-step tutorial, a blog post), controlling how information is organized and presented.
- **Audience** defines the intended reader's profile (e.g., a university student, an industry expert, a curious teenager), guiding the tone, vocabulary, and conceptual depth.

Crucially, MGA moves beyond using a small, fixed set of styles. Instead, it **adaptively generates** multiple, contextually relevant GA pairs for each source document. To achieve this, we prompted the labeler LLM to produce five distinct GA pairs and curated this data through a rigorous rule-based validation process (e.g., validating JSON structure and pair count). This filtered dataset explicitly trains the 'GA-SLM' to execute a "one-pass-for-many" strategy, mitigating the risk of mode collapse, where repeated sampling requests to a model can yield highly similar outputs.

**Stage 2: Controlled Reformulation.** This stage aims to **balance variance and invariance**, directly implementing our "Limited Consistency" principle. The core design is a finetuning strategy that, instead of narrowly optimizing for perfect outputs (e.g., a score of 5), relaxes the quality threshold to ensure a high proportion of broadly acceptable generations (a score of 3 or higher).

---

[2]Tool model training details are presented in Appendix B

To formalize this, let $D$ be a source document and $G$ be a generated GA-pair. The teacher LLM first produces an initial set of synthetic reformulations $\mathcal{D}_{\text{synth}} = \{(D_i, G_i, D_i')\}$. However, training the Tool SLM directly on this full dataset would cause it to replicate the teacher's suboptimal outputs. To circumvent this, we leverage the teacher LLM as a quality judge, using its scoring function $S(D_i') \in \{1, \ldots, 5\}$ to filter for a high-quality subset, $\mathcal{D}_{\text{SFT}}$:

$$\mathcal{D}_{\text{SFT}} = \{(D, G, D') \in \mathcal{D}_{\text{synth}} \mid S(D') \geq 3\}$$

The 'Reformulation-SLM', parameterized by $\theta$, is then trained exclusively on this curated subset using a standard supervised fine-tuning (SFT) objective:

$$\mathcal{L}_{\text{SFT}}(\theta) = \mathbb{E}_{(D,G,D') \sim \mathcal{D}_{\text{SFT}}} \left[ -\log P_\theta(D'|D, G) \right]$$

This targeted alignment process imbues the 'Reformulation-SLM' with the nuanced ability to generate novel content that remains faithful to the source material.

After the two stage synthesis, a final heuristic cleaning process is applied to the generated corpus. This stage filters out high-frequency generative patterns (e.g., '*Please note that ...*') and removes documents with extremely low keyword coverage with the source document, ensuring final data quality. Finally, we achieve a 3.9x token expansion while maintaining high quality and diversity.

## 4 EXPERIMENTS

We now empirically validate the MGA framework. We begin by establishing its core effectiveness in data-constrained scaling scenarios (Section 4.2). Following this validation, we provide a deeper analysis by addressing three key research questions (Section 4.3):

- **RQ1:** How does MGA reformulation complement existing synthetic data strategies?
- **RQ2:** What role does reformulation diversity play in high-repetition training?
- **RQ3:** Why does reformulation fundamentally benefit the model's learning process?

### 4.1 SETUP

**Models and Hyperparams**   The architecture of pretraining model follows that of Llama 3 (Dubey et al., 2024). Experiments across various sizes (134M/377M/1.7B) were running with Warmup-Stable-Decay lr scheduler (Hu et al., 2024) where 0.1% warmup steps, 75% stable and final 25% decay phase. Detailed model specifications are provided in Appendix C.2.

**Datasets**   We build MGACorpus based on SmolLM-Corpus (Ben Allal et al., 2024), which contains four subsources (fineweb-edu-dedup / cosmopedia / python-edu / open-web-math). We reformulated the 195B tokens fineweb-edu-dedup source and finally got 770B cleaned synthetic tokens.

**Evaluation**   We evaluate the models on a comprehensive suite of benchmarks include ARC-Easy / Challenge (Clark et al., 2018), HellaSwag (Zellers et al., 2019), Winogrande (Sakaguchi et al., 2020), MMLU (Hendrycks et al., 2021), GSM8K (Cobbe et al., 2021), etc., following popular practice of LIGHTEVAL (Fourrier et al., 2023) and LM-HARNESS (Gao et al., 2023). For the directly effectiveness validation of MGA, we evaluate MGACorpus aligned with Fineweb/SmolLM/Cosmopedia settings[3]. For training dynamics, we report the average of 12 benchmarks and validation losses on held-out fineweb-edu-dedup dataset.

### 4.2 MAIN EXPERIMENTS

To directly evaluate MGA's potential as a solution for data scarcity and repetition, we present a comprehensive analysis in two parts. First, we benchmark MGA's performance against recent SOTA small LMs to establish a comparative baseline. Subsequently, we investigate its behavior under data-constrained scaling scenarios, specifically situations where the training budget exceeds the available unique high-quality data, a common limitation in practical applications. [4]

---

[3]https://github.com/huggingface/cosmopedia/blob/main/evaluation
[4]Details on data recipes and comparisons with other models are provided in Appendix C.1 and D.1.

Table 2: Benchmark **MGA** with SOTA SmolLM series. Models of similar size are grouped. All results are obtained via LIGHTEVAL (Fourrier et al., 2023). The best result within each fair comparison is highlighted in green. Note that SmolLM2 models, trained with substantially more compute, are included for reference only.

| Model | #Params. | #Tokens | ARC(C+E) | Wino. | Hella. | MMLU | MMLU-PRO | CSQA | OpenBookQA | PIQA | TriviaQA | GSM8K | Avg. |
|---|---|---|---|---|---|---|---|---|---|---|---|---|---|
| SmolLM2-135M | 135M | 2T | 44.12 | 51.07 | 42.03 | 31.27 | 11.06 | 33.82 | 35 | 68.23 | 1.91 | 1.52 | 32.00 |
| SmolLM-135M | 135M | 600B | 42.47 | 51.54 | 41.08 | 29.93 | 11.4 | 32.51 | 33.2 | 68.17 | 1.08 | 0.99 | 31.24 |
| SmolLM-135M (ours) | 134M | 600B | 41.71 | 52.41 | 40.69 | 30.03 | 11.37 | 34.32 | 35.4 | 67.85 | 0.02 | 1.29 | 31.51 |
| MGA-Expansion | 134M | 600B | 43.01 | 51.7 | 41.25 | 30.1 | 11.76 | 32.68 | 36.4 | 67.3 | 2.05 | 1.44 | 31.77 |
| SmolLM2-360M | 360M | 4T | 53.4 | 52.33 | 54.58 | 35.29 | 11.17 | 37.92 | 37.6 | 71.76 | 16.73 | 2.96 | 37.37 |
| SmolLM-360M | 360M | 600B | 49.99 | 52.96 | 51.67 | 33.84 | 11.42 | 34.81 | 37.6 | 71.87 | 2.27 | 1.97 | 34.84 |
| SmolLM-360M (ours) | 377M | 600B | 48.57 | 52.64 | 51.02 | 33.63 | 11.25 | 36.77 | 39 | 71 | 0.29 | 1.52 | 34.57 |
| MGA-Expansion | 377M | 600B | 49.39 | 52.64 | 51.34 | 34.09 | 11.35 | 37.1 | 38 | 72.31 | 7.28 | 1.74 | 35.52 |
| SmolLM2-1.7B | 1.7B | 11T | 60.42 | 59.59 | 68.73 | 41.4 | 19.61 | 43.65 | 42.6 | 77.53 | 36.68 | 29.04 | 47.93 |
| SmolLM-1.7B | 1.7B | 1T | 59.95 | 54.7 | 62.83 | 39.35 | 10.92 | 38 | 42.6 | 75.9 | 13.14 | 4.62 | 40.20 |
| SmolLM-1.7B (ours) | 1.7B | 1T | 59.63 | 57.38 | 65.19 | 39.4 | 12.11 | 42.59 | 45.6 | 76.88 | 4.95 | 7.81 | 41.15 |
| MGA-Expansion | 1.7B | 1T | 60.36 | 57.46 | 65.52 | 40.79 | 14.1 | 41.11 | 42.8 | 77.53 | 20.42 | 13.87 | 43.4 |

**Performance training on MGACorpus** We evaluate whether incorporating MGA data enhances model performance compared to a baseline trained solely on the original sources, using fixed training budgets and model sizes ranging from 134M to 1.7B. As shown in Table 2, MGA-Expansion shows consistent improvements across different model sizes, with larger performance gains as model size increases, +0.26/+0.95/+2.15 for 134M/377M/1.7B models respectively. Notably, MGA-Expansion achieved substantial gains in reasoning-intensive tasks such as TriviaQA (+2.03/+6.99/+15.47) and GSM8K (+0.15/+0.22/+6.06), and shows strong performance on MMLU/MMLU-Pro. We hypothesize that MGA's data reformulation, by exposing the model to diverse phrasings of the same underlying information, fosters more robust generalization. This enhanced generalization, in turn, improves the model's reasoning capabilities, leading to the results observed on these specific benchmarks.

**Scaling Dynamics** We further investigate MGA's behavior under data-constrained scaling scenarios. Models of 377M/1.7B/7B/13B are trained using a learning rate scheduler with only warmup and stable phases, which allows for a direct performance comparison across repetition epochs.

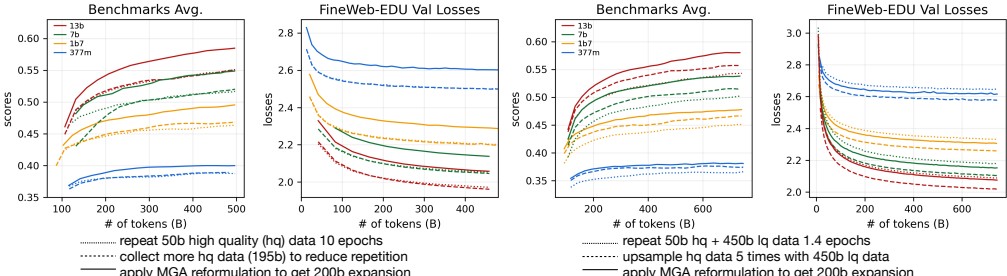

Figure 3: Training dynamics of two common scenarios under data-constrained conditions: (1) expanding a 50B high-quality dataset to a 500B training budget (entire set repetition), (2) expanding a 500B mixed-quality dataset to a 700B training budget (subset repetition). Data recipe and benchmark details are provided in Appendix C.1 and C.3.

**Scaling Results** As shown in Figure 3, MGA demonstrates favorable scaling properties with both data budget (D-scaling) and model sizes (N-scaling) under two common data-constrained scenarios.

- In the entire set experiments, simply increasing unique token count by collecting more high quality data (195B via Full-Fineweb-Edu) shows marginal improvements (+0.2/+0.15/-0.16/+0.11) at 200/300/400/500 billion token steps (13B size). In contrast, MGA, through a 200B reformulation as expansion of the original 50B data, demonstrates consistent gains (+2.65/+3.14/+3.43/+3.46), highlighting **effective D-scaling**.

- Similarly, in the subset experiments, both upsampling the high-quality sub data portion (5x) and MGA (via a 200B expansion) improve upon the baseline. However, their N-scaling properties with model parameters differ significantly: the performance advantage of upsampling remains relatively constant across model sizes (+0.89/+1.53/+1.23/+1.41), whereas MGA ex-

pansion exhibits **superior N-scaling**, its performance gains amplifying with increasing model scale (+1.46/+2.67/+3.59/+3.73).

These scaling experiments confirm that MGA is a powerful data augmentation strategy that aids both model (N) and data (D) scaling in constrained scenarios. Notably, MGA's performance advantage emerges from the very first epoch, well before significant data repetition occurs, and this gap widens as training progresses. The dual observation leads directly to our core research questions, which we will investigate in Section 4.3: How does MGA's inherent diversity mitigate the degradation from high-repetition training (**RQ2**), and what fundamental learning benefits does the reformulation provide from the outset (**RQ3**)?

**Validation Losses** Although MGA demonstrates superior benchmark performance, we observe increasing validation losses compared to baseline models. While higher validation losses might seem concerning at first glance, it's important to note that validation loss may not fully reflect model performance, as token-level perplexity is inherently biased by the frequency distribution of the validation set, and in-domain validation metrics may not necessarily correlate with out-of-domain generalization capabilities. This observation, combined with recent studies linking loss degradation to model collapse (Dohmatob et al., 2024b;a; Zhu et al., 2024), calls for a more nuanced analysis, which we will also provide in Section 4.3.3.

## 4.3 DISCUSSIONS

The strong empirical results in our main experiments validate MGA as a powerful data augmentation strategy. However, these outcomes naturally lead to deeper questions about its positioning, mechanics, and underlying benefits. In the following section, we move from empirical validation to analytical discussion, addressing the key research questions outlined at the beginning of this section.

### 4.3.1 HOW DOES MGA COMPLEMENT EXISTING SYNTHETIC DATA STRATEGIES?

Our main experiments demonstrated MGA's value relative to data repetition, but how does it stand within the diverse and rapidly evolving ecosystem of synthetic data? To address RQ1, this section positions MGA not as a standalone replacement, but as a complementary approach to other prevalent strategies, such as Nemotron-CC-Synthetic (Su et al., 2024). We compare MGA-enhanced data with this popular open-source synthetic datasets to highlight its unique contribution.

To conduct a fair comparison, we designed a controlled experiment with four distinct data mixtures, each training a 1.7B parameter model for 800B tokens. The data blend setups were as follows:

- Baseline: A high-quality real dataset fineweb-edu.
- Exp A: 35% token budget replaced by Nemotron-CC-HQ synthetic corpus (+Nemotron-Syn).
- Exp B: 35% token budget replaced by MGACorpus (+MGA).
- Exp C: 70% token budget replaced by an equal combination of Nemotron-Syn and MGACorpus data (+Nemotron-Syn +MGA).

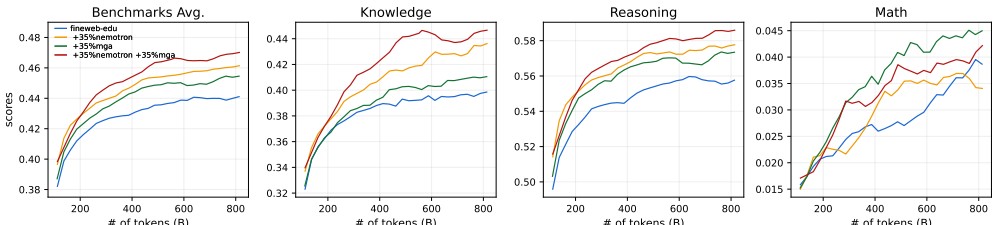

Figure 4: Benchmark results demonstrating the complementary nature of different synthetic data strategies. While both MGA and Nemotron-Syn individually improve over the baseline, their combination (Exp C) yields a significant synergistic boost in performance. For a detailed breakdown of performance by each synthetic task, as well as supplementary cross-mixing experiments that further validate MGA's role in creating a generalizable base model, please refer to Appendix D.3.

As illustrated in Figure 4, the results reveal a clear performance hierarchy: Exp C > Exp A > Exp B > Baseline. The strong performance of Exp A is understandable, as Nemotron-Syn is a high-quality and diverse synthetic corpus composed of five diverse subsets. The inclusion of various

data formats, some of which (like QA pairs) align well with common evaluation structures. While our MGA-enhanced mix (Exp B) also surpasses the baseline, the most compelling finding comes from Exp C, which significantly outperforms all other configurations. This demonstrates a clear synergistic effect, where the structural and stylistic diversity from MGA's reformulation enriches the high-quality, task-aligned data from Nemotron-Syn.

Therefore, we answer RQ1 by concluding that MGA is not in competition with but is complementary to other synthetic data methodologies. The path to resolving data scarcity does not lie in a single synthesis technique, but in the thoughtful combination of diverse strategies. MGA provides a foundational, general-purpose enhancement through reformulation that benefits even further when combined with specialized, task-aligned synthetic data.

### 4.3.2 DOES REFORMULATION DIVERSITY HELP TO MITIGATE REPETITION ISSUE?

To address **RQ2**, this section examines how different design choices in prompt engineering influence the effectiveness of the MGA framework, particularly under high-repetition conditions. By comparing SLM variants (introduced in Section 3.1) using different consistency requirements (see Table 3), we identify optimal strategies for balancing information preservation with content diversity.

Table 3: Performance comparison of different SLM variants on reformulation quality metrics.

| Models | Total Examples | 5 | 4 | 3 | 2 | 1 | Rate($\leq 2$) | Rate($\geq 4$) | Rate($= 5$) |
|---|---|---|---|---|---|---|---|---|---|
| SLM-Base | 15,355 | 3,788 | 7,124 | 3,224 | 736 | 285 | 6.65% | 71.06% | 24.67% |
| SLM-Strict | 15,355 | 6,814 | 5,220 | 2,384 | 520 | 227 | 4.86% | **78.37%** | **44.38%** |
| SLM-Relaxed | 15,355 | 408 | 1,685 | 3,889 | 4,156 | 5,086 | **60.19%** | 13.63% | 2.66% |

We sample an additional 20B tokens from real data and generate three synthetic datasets: 80B tokens using SLM-Base, 80B tokens using SLM-Strict, and 40B tokens using SLM-Relaxed. As mentioned before, SLM-Base expands the original corpus to 3.9× more tokens, while SLM-Relaxed makes only 2× tokens as we only require basic topical relevance. Similar to experimental setup in early sections, we set a high-repetition baseline on a smaller data scale (replicating the original 20B tokens 10 times) to more clearly demonstrate the potential impact of SLM-Strict compared to SLM-Base in high-repetition scenarios.

As shown in Figure 5, our experiments reveal distinct patterns across training configurations. Both SLM-Base and SLM-Strict show performance improvements, while the SLM-Relaxed configuration leads to significant collapse. More supplementary experiments could be found in Appendix D.2.

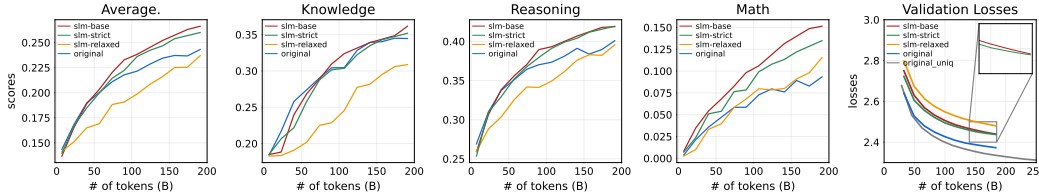

Figure 5: Benchmark results and validation losses. The sensitivity to data repetition varies across capability domains, with knowledge dimension showing greater resilience.

Despite the apparent effectiveness of strict information preservation, can it fundamentally address the challenges posed by data repetition? Our examination of validation loss trajectories reveals a critical distinction: SLM-Base maintains healthy optimization characteristics throughout training, whereas SLM-Strict exhibits degraded scaling behavior at higher iteration steps, reminiscent of the limitations observed with data repetition.

Therefore, this investigation into different prompt engineering strategies concludes that a balanced 'Limited Consistency' approach (SLM-Base) yields the best reformulation quality and subsequent model performance answering to **RQ2**.

### 4.3.3 WHY DOES REFORMULATION BENEFIT THE MODEL'S LEARNING PROCESS?

Having explored the impact of diversity in addressing data repetition (**RQ2**), we now turn to **RQ3**: this section investigates the underlying mechanisms by analyzing learning characteristics and validating against potential issues like model collapse (Dohmatob et al., 2024b;a; Zhu et al., 2024).

**Multi-perspective Validation Analysis**   Our analyses across different validation sets reveal varying patterns in model behavior (Figure 6). As expected, MGA groups' substitution of fineweb-edu data results in adverse effects on corresponding loss, with similar deterioration observed in open-web-math. Interestingly, the synthetic dataset cosmopedia demonstrates improved loss metrics. A notable contrast emerges in python-edu: while MGA exhibit negative impact at the 134M and 377M parameter, this trend reverses at 1.7B, suggesting scale-dependent effects on model behavior.

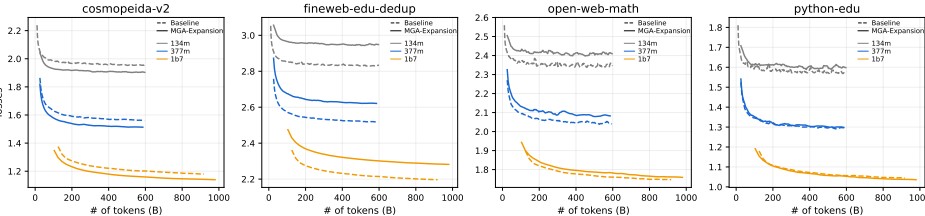

Figure 6: validation losses of experiments in Section 4.2.

**Fine-grained Pattern Analysis**   To better understand whether increased validation loss truly indicates model collapse, we conduct a fine-grained analysis of loss patterns. Specifically, we compare token-level losses of 800B checkpoint between models trained on real data and synthetic data (Baseline and MGA-Expansion in Section 4.2, respectively). The document samples are from both Fineweb-Edu and MGACorpus. As illustrated in subfigures 1 and 3 of Figure 7, each point represents a sample's average token loss, consistent with the overall loss discrepancy shown in Figure 6.

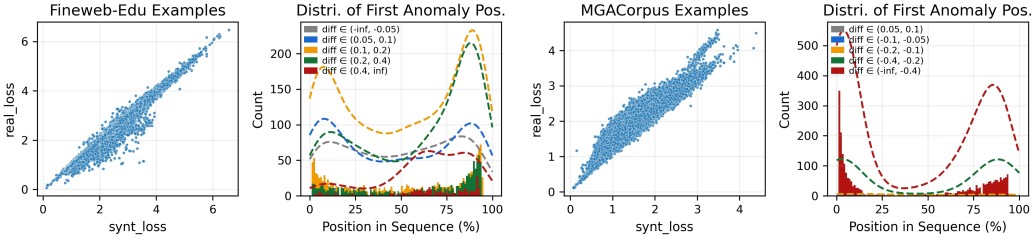

Figure 7: Losses pattern analysis. Subfigures 1 and 3 shows comparison between models trained on different data settings, with $loss_{real}$ on y-axis and $loss_{synt}$ on x-axis. Subfigures 2 and 4 track the position where $loss_{synt}^i - loss_{real}^i$ ($loss_{diff}^i$) first becomes significantly higher than the sequence's average difference (detailed definition in Appendix D.4).

The distribution of first anomaly positions (subfigures 2 and 4) reveals a crucial insight: when processing real data, models trained on synthetic data show performance degradation (measured by $loss_{diff}$) that predominantly manifests in later sequence positions, which intensifies as $loss_{diff}$ increases. However, this positional bias disappears when evaluating on synthetic data.

The systematic pattern suggests that rather than experiencing model collapse, the synthetic-trained model may have developed a different learning strategy (examples shown in Appendix D.4). While it shows higher validation losses on certain real-world datasets, its strong performance in our main experiments indicates a potential trade-off: the model may prioritize learning generalizable patterns from context over memorizing specific sequence dependencies. This shift in learning process could explain both the improved performance on benchmark tasks and the increased losses on validation sets that potentially require more memorization-based processing.

These findings indicate that the performance characteristics associated with MGA data likely stem from altered learning strategies, potentially prioritizing generalizability, rather than representing model collapse, which addressing **RQ3**,

## 5   CONCLUSION

In this work, we introduced MGA, a principled framework that leverages genre-audience reformulation to systematically expand and augment existing corpora with diverse, synthetically generated variations. Our core finding highlights MGA's effectiveness as a data augmentation strategy specifically targeting the repetition challenge: in data-constrained scaling experiments, MGA significantly

outperformed naive data repetition and simple upsampling, enabling more effective model training beyond unique data limits. Furthermore, the quality of the MGA was confirmed by consistent performance improvements when incorporated into standard training mixtures across various model sizes. Crucially, our experiments also revealed MGA's role as a complementary strategy, demonstrating a powerful synergistic effect when combined with other prominent synthetic datasets. In essence, MGA's effectiveness demonstrates that the key to overcoming data limits is generating relevant diversity, not just raw volume. Therefore, our work offers more than a tool; it provides a new roadmap for the community, where the thoughtful combination of different approaches becomes the cornerstone of sustainable progress in the continue scaling of LLM development.

## REPRODUCIBILITY STATEMENT

We are committed to reproducibility and will release the 770B-token MGACorpus and all associated resources. Our MGA framework implementation, including tool model details, is documented in Appendix B. The complete pretraining and evaluation setup, with data recipes and hyperparameters, is detailed in Appendix C. Cases and prompts used to curate the dataset is provided in Appendix E.

## ACKNOWLEDGEMENTS

We are grateful to Chao He, Zhixin Yao, Yue Chen and Runyu Shi for their help with prompt templates and case studies, and to Seed-Foundation team for providing the stable training/inference platform, which enabled us to build the synthetic pipeline and corpus within a reasonable timeframe.

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

## A    LIMITATIONS AND FUTURE WORK

While our experimental results demonstrate the effectiveness of MGA in both quality validation and scaling scenarios, several important aspects warrant further investigation. We identify two key areas for future research:

- Our current experiments demonstrate effectiveness up to 13B parameters and 1,000B tokens of training budget. Extending this approach to long-horizon training and larger-scale models requires additional validations, particularly for next-generation models which require hundreds of trillions of training tokens.

- Regarding data repetition strategies, we present preliminary explorations under computational resource constraints. The underlying patterns and their sensitivity to various factors, such as repetition ratio, data distribution, and even model hyperparameters, require systematic investigation. Future research should examine how these factors collectively determine optimal data strategies across different training scenarios.

**Broader Impact**    This paper explores the use of LLMs as a data expansion method for pretraining large language models. We introduce the MGAframework to mitigate data repetition issues, which holds potential for positive societal impact, particularly in synthetic data generation for training language models. Nonetheless, the use of synthetic data generated by LLMs is not without risks; for instance, LLM hallucinations, even after filtering, could introduce novel errors or biases into models trained on such data, a factor that warrants careful consideration in future research and deployment.

## B    MGA FRAMEWORK IMPLEMENTATION DETAILS

### B.1    TOOL MODEL TRAINING & RESOURCES

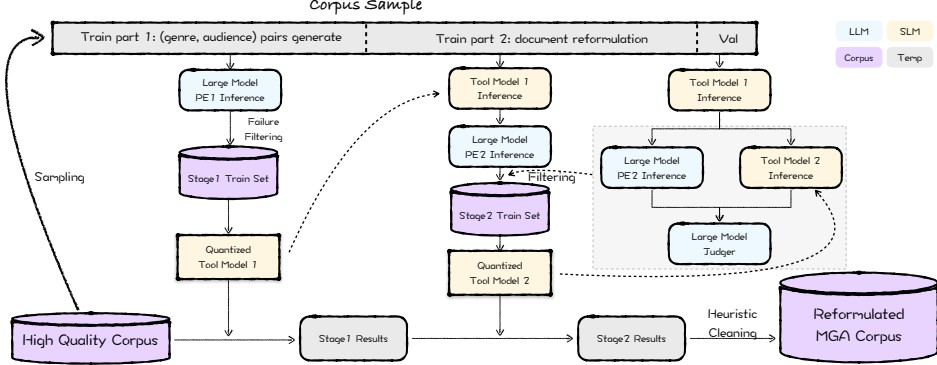

Figure 8: Implementation details. From a high-quality corpus, we sample a subset to serve as input for the LLM labeler and judger. Through data filtering, we train and quantize tool SLMs for each stage to improve inference efficiency, which are used to generate the reformulated corpus.

**High Quality Corpus**    We conduct our reformulated corpus based on SmolLM-Corpus[5] (Ben Allal et al., 2024), expanding fineweb-edu-dedup source from 195B tokens to 770B tokens. Then we setup additionally experiments on FineWeb and FineWeb-Edu (Penedo et al., 2024), which constitute a solid foundation for research on data scaling approaches. Prior to these experiments, we have validated our approach on our in-house datasets. The results demonstrate consistent performance across both datasets, suggesting broad applicability of our method.

---

[5]https://github.com/huggingface/smollm/tree/main/text/pretraining

**Tool Models Training** Initialized from a pretrained SLM (a 3.3B MoE model), we collect 50,000 training samples through LLM teacher, where 15,000 of raw text to genre-audience pairs, 35,000 of raw text to reformulated output. Each model's validation responses are scored by capable LLM judger, that ensures the SLMs achieve comparable synthesis quality to the LLM labeler as shown in Table 1. The sequence length is 8192 with maximum prompt/response length 4,096 tokens, each model is trained 3 epochs on the samples with a cosine lr scheduler.

We also finetuned a public model, Qwen3-30B-A3B Base(Yang et al., 2025), to serve as the 'Open Tool SLM', which will also be released. The results confirm that its performance is highly comparable to our internal models, proving our method is not dependent on any specific proprietary tool, as shown in Table 4. The result provides conclusive evidence that our core methodology is a robust and reproducible technique that can be readily implemented by other researchers using public models.

Table 4: Reformulation quality comparison between finetuned qwen3-a3b and its LLM teacher.

| Models | Total Examples | 5 | 4 | 3 | 2 | 1 | Rate($\geq 3$) | Diff |
|---|---|---|---|---|---|---|---|---|
| Labeler LLM | 15,355 | 4,120 | 7,143 | 3,034 | 661 | 214 | 93.11% | - |
| Tool SLM | 15,355 | 3,788 | 7,124 | 3,224 | 736 | 285 | 92.06% | -1.05% |
| Qwen3-30B-A3B (Finetuned) | 15,355 | 3,322 | 7,154 | 3,614 | 805 | 249 | 91.76% | -1.35% |

**Quantifying Generation Diversity** To systematically validate our design choices in preventing mode collapse and ensuring synthetic data heterogeneity, we introduce two metrics: Reference Similarity (RefSim), which measures the average cosine similarity between synthetic documents and their original sources, and Heterogeneity Score, defined as $1 - \frac{1}{|D|(|D|-1)} \sum_{i \neq j} \mathrm{sim}(\mathrm{pair}_i, \mathrm{pair}_j)$ to evaluate internal corpus diversity.

As shown in Table 5, our base MGA strategy strikes an optimal balance. While a relaxed prompt strategy (free LLM sampling) leads to severe homogenization (lowest Heterogeneity Score), and a strict prompt policy limits diversity to below that of the original data, our approach maximizes internal diversity while preserving relevance to the source text.

Table 5: Trade-off between source relevance and internal diversity across different prompt strategies.

| Setting (Prompt Strategy) | RefSim (vs. Original) | Heterogeneity Score |
|---|---|---|
| Original Data | 1.0000 | 0.3355 |
| **Base (Our Method)** | **0.8979** | **0.3390** |
| Strict (Enforce high similarity) | 0.9392 | 0.3326 |
| Relaxed (LLM free-sampling) | 0.8319 | 0.3063 |

Furthermore, we evaluate the impact of our "one-pass-for-many" generation strategy against a standard "one-pass-for-one" baseline. As detailed in Table 6, rather than introducing conditioning bias, our strategy acts as a diversity catalyst. It yields a significantly higher number of unique genre-audience pairs and achieves superior lexical and semantic diversity, confirming its effectiveness in scaling data synthesis without collapsing into repetitive patterns.

Table 6: Diversity metrics comparing our "one-pass-for-many" strategy against standard sampling.

| Metric | One-pass-for-many (Ours) | One-pass-for-one |
|---|---|---|
| Total Examples | 1000 | 1000 |
| Unique Genres | **913** | 779 |
| Unique Audiences | **871** | 715 |
| Unique Pairs | **987** | 951 |
| Type-Token Ratio (TTR) | **0.1213** | 0.0839 |
| TF-IDF Dissimilarity | **0.9751** | 0.9572 |

**Resource Analysis** To generate 770B synthetic tokens, it takes 256×64 and 1024×130 NVIDIA H100 GPU hours to process two stages.

## C PRETRAINING AND EVALUATION SETUP

### C.1 PRETRAINING DATA RECIPES

**Data Recipe** The training token budgets are 600B/600B/1000B for size of 134M/377M/1.7B models, which are aligned with SmolLM1 series (Ben Allal et al., 2024). Our baseline is trained on SmolLM-Corpus dataset, in contrast to SmolLM's recipe, we use unique token number from each source as the mixing ratio shown in Table 7. This ensures that different sources have consistent repetition epochs during training. For a fair comparison, the mixing ratios of other data sources are kept constant across experiments. We specifically adjusted the proportions of fineweb-edu-dedup and MGACorpus to isolate the impact of the MGA reformulation.

Table 7: MGACorpus experiments data recipe: source weight (%) and #unique_tokens × #epochs.

| experiments | - | fineweb-edu-dedup | cosmopedia-v2 | python-edu | open-web-math | MGACorpus |
|---|---|---|---|---|---|---|
| Baseline | weight | 80.89 | 11.65 | 1.66 | 5.80 | - |
|  | #unique_tokens × #epochs | 195 × 4.15 | 28 × 4.15 | 4 × 4.15 | 14 × 4.15 | - |
| MGA-Expansion | weight | **16.29** | 11.65 | 1.66 | 5.80 | **64.59** |
|  | #unique_tokens × #epochs | 195 × 0.84 | 28 × 4.15 | 4 × 4.15 | 14 × 4.15 | 770 × 0.84 |

The experiment design for different strategies is presented in Table 8, which involves three datasets: (1) a 50B-token random sample from fineweb-edu-dedup, (2) a corresponding filtered subset from MGACorpus, and (3) a 450B-token deduplicated corpus obtained from Fineweb.

Table 8: Scaling experiments data recipe, values represent #unique_tokens × #epochs.

| Repetition | Experiments | Training Budget | fineweb-edu dedup | MGA corpus | fineweb random | Design Rationale |
|---|---|---|---|---|---|---|
| EntireSet | Baseline | 500B | 50 × 10 | - | - |  |
|  | Full-Fineweb-Edu | 500B | 195 × 2.56 | - | - | What if we could collect more unique data. |
|  | MGA Expansion | 500B | 50 × 2 | 200 × 2 | - | Add MGA to reduce the repetition num. |
| Subset | Baseline | 700B | 50 × 1.4 | - | 450 × 1.4 |  |
|  | Upsample-EDU | 700B | 50 × 5 | - | 450 × 1 | Upsample to get 200B more budget. |
|  | MGA Expansion | 700B | 50 × 1 | 200 × 1 | 450 × 1 | Add MGA to achieve the same target. |

### C.2 MODEL HYPERPARAMETERS

We sample 100 million tokens from SmolLM-Corpus as the validation dataset. The hyperparams are presented in Table 9. These hyperparameters are determined by scaling laws to ensure an optimal baseline and are kept consistent across all experimental groups. The tokenizer used for training and computing token counts is the same as SmolLM1[6] with a vocab size of 49,152.

Table 9: Hyperparams of different model size.

| model size | batch size | learning rate | weight decay | hidden size | ffn inner | num heads | num layers | shared q_head | seq len | tie emb | total params |
|---|---|---|---|---|---|---|---|---|---|---|---|
| **134M** | 128 | 3e-3 | 0.1 | 1,204 | 4,096 | 8 | 8 | 1 | 8,192 | false | 134M |
| **377M** | 320 | 1.5e-3 | 0.1 | 1,536 | 6,144 | 12 | 10 | 1 | 8,192 | false | 377M |
| **1.7B** | 512 | 5e-4 | 0.1 | 2,560 | 10,240 | 20 | 16 | 1 | 8,192 | false | 1.68B |
| **7B** | 1,024 | 4e-4 | 0.1 | 4,096 | 8,192 | 32 | 32 | 4 | 8,192 | false | 6.98B |
| **13B** | 1,024 | 4e-4 | 0.1 | 4,096 | 12,288 | 32 | 48 | 4 | 8,192 | false | 12.9B |

### C.3 EVALUATION DETAILS

The LightEval results provided in Section 4.2 follow SmolLM setting, that with GSM8K/MMLU 5-shot and all the others 0-shot. The benchmarks presented in Figure 9 and Figure 10 follow few-shot evaluation settings, specifically ARC(8-shots), TriviaQA(5-shots), Winogrande(5-shots) and similar configurations for other tasks.

---

[6]https://huggingface.co/HuggingFaceTB/cosmo2-tokenizer

# D  SUPPLEMENTARY EXPERIMENTAL RESULTS & ANALYSIS

## D.1  BENCHMARK COMPARISONS WITH MORE SOTA MODELS

While model performance is influenced by multiple factors, we list some recently SOTA small language models as reference.

Table 10: Benchmark **MGA** with SOTA small LMs. Models of similar size are grouped. All results are obtained through LIGHTEVAL (Fourrier et al., 2023). Best results in each group are highlighted in **bold**, the second in underline, and in green for that MGA wins under fair comparison.

| Model | #Params. | #Tokens | ARC(C+E) | Wino. | Hella. | MMLU | MMLU-PRO | CSQA | OpenBookQA | PIQA | TriviaQA | GSM8K | Avg. |
|---|---|---|---|---|---|---|---|---|---|---|---|---|---|
| SmolLM2-135M | 135M | 2T | **44.12** | 51.07 | **42.03** | **31.27** | 11.06 | 33.82 | 35 | **68.23** | 1.91 | **1.52** | **32.00** |
| SmolLM-135M | 135M | 600B | 42.47 | 51.54 | 41.08 | 29.93 | 11.4 | 32.51 | 33.2 | 68.17 | 1.08 | 0.99 | 31.24 |
| SmolLM-135M (ours) | 134M | 600B | 41.71 | **52.41** | 40.69 | 30.03 | 11.37 | **34.32** | 35.4 | 67.85 | 0.02 | 1.29 | 31.51 |
| MGA-Expansion | 134M | 600B | 43.01 | 51.7 | 41.25 | 30.1 | 11.76 | 32.68 | **36.4** | 67.3 | **2.05** | 1.44 | 31.77 |
| Qwen2.5-0.5B | 360M | 18T | 45.16 | **53.99** | 51.16 | 33.51 | **11.97** | 31.61 | 37.6 | 69.97 | 3.96 | **32.9** | 37.18 |
| SmolLM2-360M | 360M | 4T | **53.4** | 52.33 | **54.58** | **35.29** | 11.17 | **37.92** | 37.6 | 71.76 | **16.73** | 2.96 | 37.37 |
| SmolLM-360M | 360M | 600B | 49.99 | 52.96 | 51.67 | 33.84 | 11.42 | 34.81 | 37.6 | 71.87 | 2.27 | 1.97 | 34.84 |
| SmolLM-360M (ours) | 377M | 600B | 48.57 | 52.64 | 51.02 | 33.63 | 11.25 | 36.77 | **39** | 71 | 0.29 | 1.52 | 34.57 |
| MGA-Expansion | 377M | 600B | 49.39 | 52.64 | 51.34 | 34.09 | 11.35 | 37.1 | 38 | **72.31** | 7.28 | 1.74 | 35.52 |
| Qwen2.5-1.5B | 1.3B | 18T | 58.36 | 58.64 | 66.39 | 40.23 | 13.85 | 34.4 | 39.6 | 75.95 | 20.51 | **60.8** | 46.87 |
| SmolLM2-1.7B | 1.7B | 11T | **60.42** | **59.59** | **68.73** | **41.4** | **19.61** | **43.65** | 42.6 | **77.53** | **36.68** | 29.04 | 47.93 |
| Llama-3.2-1B | 1.2B | 9T | 49.2 | 57.8 | 61.2 | 36.63 | 11.7 | 41.2 | 38.4 | 74.8 | 28.1 | 7.2 | 40.62 |
| OLMo-1B-0724 | 1B | 3.05T | 44.71 | 56.04 | 64.38 | 32.3 | 11.8 | 33.09 | 38 | 75.24 | 13.82 | 2.43 | 37.18 |
| SmolLM-1.7B | 1.7B | 1T | 59.95 | 54.7 | 62.83 | 39.35 | 10.92 | 38 | 42.6 | 75.9 | 13.14 | 4.62 | 40.20 |
| SmolLM-1.7B (ours) | 1.7B | 1T | 59.63 | 57.38 | 65.19 | 39.4 | 12.11 | 42.59 | **45.6** | 76.88 | 4.95 | 7.81 | 41.15 |
| MGA-Expansion | 1.7B | 1T | 60.36 | 57.46 | 65.52 | 40.79 | 14.1 | 41.11 | 42.8 | 77.53 | 20.42 | 13.87 | 43.4 |

In our experimental observations (Table 10), notable performance improvements are demonstrated in both TriviaQA and GSM8k benchmarks, warranting a detailed examination of these score variations. The enhanced TriviaQA performance exhibited by SmolLM1-1.7B relative to our baseline can be attributed to the larger proportion of Cosmopedia in its training configuration. Both MGACorpus and Cosmopedia employ synthetic methodologies, which contribute to improved learning efficiency. The observed gains in GSM8K performance can be traced to the target genres, including teaching schemas and problem-solving exemplars, embedded within the Reformulation component. This early exposure to structured problem-solving approaches facilitates more effective performance on analogous mathematical reasoning tasks.

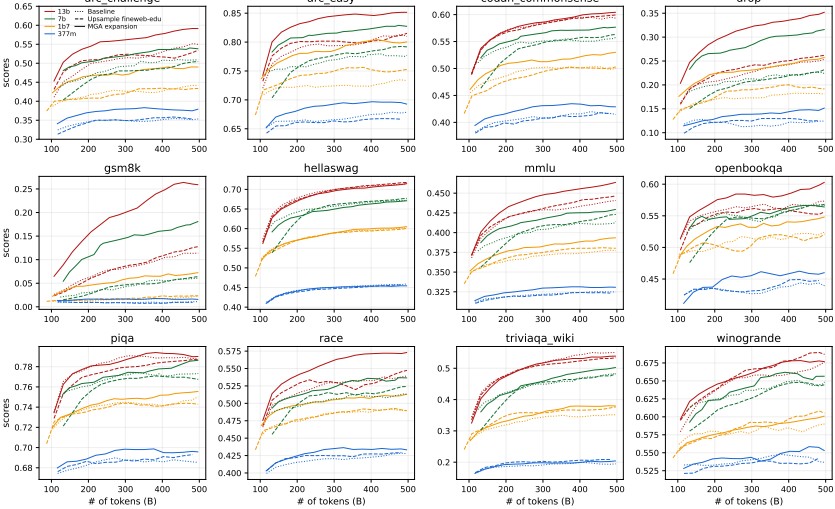

Figure 9: Detail evaluation results of EntireSet described in Table 8. MGACorpus group demonstrats advantages over other groups across most evaluation sets, consistently across models of sizes.

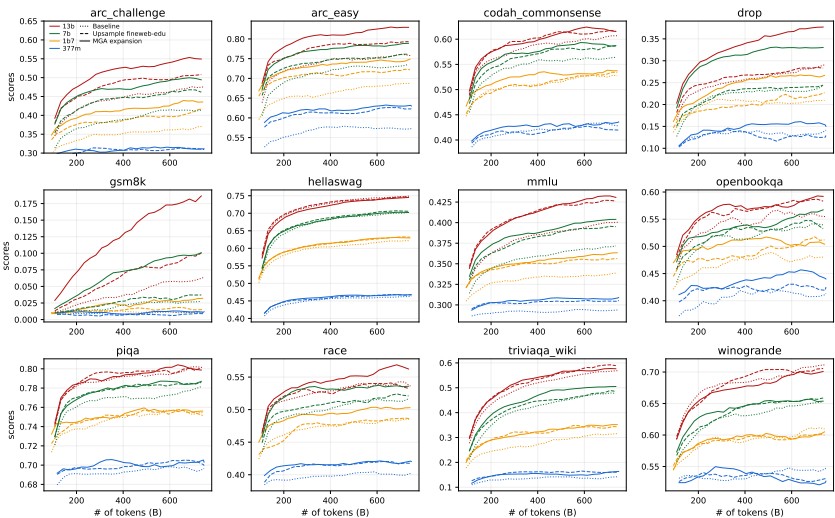

Figure 10: Detail evaluation results of Subset described in Table 8. As the model size increases, the performance gap between the upsampling group and MGACorpus gradually widens in ARC, DROP, GSM8K, RACE, but with some variations observed in TriviaQA and WinoGrande.

## D.2 "MGA-ONLY" EXPERIMENT

Our primary goal with MGA is efficient dataset expansion, typically achieved by mixing the generated corpus with existing real data, aligning with current best practices for leveraging synthetic data. However, to better characterize the properties of the MGACorpus itself and understand the impact of training exclusively on reformulated content, we also investigate an experimental setting where MGACorpus completely replaces its source data (fineweb-edu-dedup).

Table 11: MGACorpus experiments data source weight (%).

| experiments | fineweb-edu-dedup | cosmopedia-v2 | python-edu | open-web-math | MGA-corpus |
|---|---|---|---|---|---|
| Baseline | 80.89 | 11.65 | 1.66 | 5.80 | - |
| MGA-Expansion | **16.29** | 11.65 | 1.66 | 5.80 | **64.59** |
| MGA-Only | - | 11.65 | 1.66 | 5.80 | **80.89** |

As shown in Table 12, the absence of real data leads to performance degradation across most tasks (average -0.95), particularly in two tasks, Hellaswag(-1.23/-1.69/-2.85) and CommonsenseQA(-3.11/-4.83/-4.50). This decline can be attributed to our design choice, which focuses on diversity and overall quality rather than requiring the preservation of all information from each raw documents.

Table 12: Comparison between MGA-Expansion and MGA-Only

| Model | #Params. | #Tokens | ARC(C+E) | Wino. | Hella. | MMLU | MMLU-PRO | CSQA | OpenBookQA | PIQA | TriviaQA | GSM8K | Avg. |
|---|---|---|---|---|---|---|---|---|---|---|---|---|---|
| MGA-Expansion | 134M | 600B | **43.01** | **51.7** | **41.25** | **30.1** | **11.76** | **32.68** | **36.4** | 67.3 | 2.05 | **1.44** | **31.77** |
| MGA-Only | 134M | 600B | 41.98 | 51.38 | 40.02 | 29.87 | 11.5 | 29.57 | 33 | **68.01** | **2.26** | 1.06 | 30.87 |
| | | | ↓-1.03 | ↓-0.32 | ↓-1.23 | ↓-0.23 | ↓-0.26 | ↓-3.11 | ↓-3.40 | ↑0.71 | ↑0.21 | ↓-0.38 | ↓-0.90 |
| MGA-Expansion | 377M | 600B | **49.39** | 52.64 | **51.34** | **34.09** | 11.35 | **37.1** | 38 | **72.31** | 7.28 | **1.74** | **35.52** |
| MGA-Only | 377M | 600B | 47.95 | **53.35** | 49.65 | 33.31 | **11.38** | 32.27 | 38 | 70.95 | 6.83 | 1.59 | 34.53 |
| | | | ↓-1.44 | ↑0.71 | ↓-1.69 | ↓-0.78 | ↑0.03 | ↓-4.83 | - | ↓-1.36 | ↓-0.45 | ↓-0.15 | ↓-0.99 |
| MGA-Expansion | 1.7B | 1T | **60.36** | **57.46** | **65.52** | **40.79** | **14.1** | **41.11** | 42.8 | **77.53** | **20.42** | **13.87** | **43.40** |
| MGA-Only | 1.7B | 1T | 59.02 | 57.06 | 62.67 | 40.34 | 13.51 | 36.61 | **45.2** | 76.71 | 19.78 | 13.57 | 42.45 |
| | | | ↓-1.34 | ↓-0.40 | ↓-2.85 | ↓-0.45 | ↓-0.59 | ↓-4.50 | ↑2.40 | ↓-0.82 | ↓-0.64 | ↓-0.30 | ↓-0.95 |

**MGA-Only Setting of PE Ablation**   Upon relaxing the information preservation requirements for PE objectives in the MGA-Only setting, we observe a complete collapse in knowledge-based dimensions while maintaining modest improvements in reasoning and mathematical capabilities. This divergence suggests that different cognitive capabilities have distinct requirements for the richness and nature of training data content.

Figure 11: Corresponding benchmark results described in Section 4.3.2.

## D.3 COMPARISON WITH OPEN SYNTHETIC DATA

This section provides crucial empirical support for the central thesis of our Discussion (Section 4.3): the path to resolving data scarcity lies not in a single synthesis technique, but in the thoughtful combination of diverse strategies. We argue for an approach that breaks down the walls between **different synthetic data paradigms**, demonstrating that a model's robustness comes from being trained on a rich variety of data formats. The following experiments are designed to validate this philosophy, showing how MGA acts as a synergistic component within a broader data ecosystem, rather than a standalone solution.

### D.3.1 HEAD-TO-HEAD PERFORMANCE AGAINST OPEN SYNTHETIC DATASETS

This first experiment aims to benchmark MGA's effectiveness against other well-known synthetic data generation methods when each is used as the primary training corpus. We trained 377M parameter models for 300B tokens on several distinct datasets to establish a clear performance baseline. For a fair comparison with Cosmopedia, the MGA corpus was sampled to 28B unique tokens, and both datasets were repeated approximately 10.7 times during training.

Table 13: Comparative benchmark performance of 377M models trained on MGA reformulations versus other synthetic datasets for 300B tokens. For a fair comparison with Cosmopedia, MGA is sampled to 28B unique tokens, with both datasets then repeated 10.7 times during training. All benchmarks are 0-shot evaluations (obtained through LIGHTEVAL), except for MMLU (5-shot).

| Category | Document Sources | Synthetic Target | ARC(C+E) | Wino. | Hella. | MMLU | CSQA | OpenBookQA | PIQA | TriviaQA | Avg. |
|---|---|---|---|---|---|---|---|---|---|---|---|
| Cosmopedia | Textbooks/Webs | Story/Textbook/Wiki mix | 42.15 | 50.43 | 45.06 | 29.17 | 30.38 | 33.2 | 68.77 | 0.23 | 35.57 |
| MGA | High quality webs | Diverse Genre-Audience | 45.65 | 51.22 | 42.31 | 31.42 | 32.19 | 37.2 | 68.39 | 3.79 | 37.28 |
| | Low quality webs | Wrap-medium (Wiki style) | 29.01 | 50.83 | 38.36 | 26.29 | 29.32 | 32 | 67.03 | 0 | 31.72 |
| | | Extract knowledge | 40.42 | 53.2 | 44.65 | 30.57 | 28.99 | 35 | 69.42 | 0.96 | 35.72 |
| Nemotron-CC | High quality webs | Knowledge list | 42.08 | 52.17 | 42.7 | 30.71 | 32.51 | 35.4 | 70.08 | 0 | 36.21 |
| | | Concise and clear passage | 42.22 | 52.01 | 43.99 | 30.96 | 31.53 | 35 | 69.7 | 0.06 | 36.21 |
| | | Wrap-medium (Wiki style) | 42.95 | 52.17 | 43.72 | 31.06 | 31.53 | 36.2 | 70.13 | 0.82 | 36.63 |
| | | Diverse QA pairs | 46.96 | 52.57 | 49.03 | 31.36 | 38.82 | 38.8 | 70.84 | 9.21 | 40.72[7] |
| MGA | High quality webs | Diverse Genre-Audience | 45.33 | 52.41 | 42.42 | 31.33 | 31.45 | 38 | 68.61 | 4.24 | 37.34 |

The results in Table 13 highlight MGA's competitive performance as a general-purpose augmentation strategy. MGA (average 37.28) surpasses Cosmopedia (35.57), which is a blend of story, textbook, and wiki formats. When compared against the various synthesis strategies from Nemotron-CC, MGA (average 37.34) again shows strong performance, outperforming most alternatives such as 'extract knowledge' (35.72) and 'wrap-medium (Wiki style)' (36.63).

Notably, while Nemotron's diverse QA slice achieves the highest average score (40.72), this advantage is likely attributable to its format aligning directly with the question-answering structure prevalent in our 0-shot evaluation benchmarks. Despite this format-specific advantage, MGA's broader reformulation approach proves its robust utility by outperforming five of the six Nemotron strategies. This underscores MGA's value in building a well-rounded and capable base model.

### D.3.2 INVESTIGATING SYNERGISTIC EFFECTS VIA CONTINUED PRE-TRAINING

Moving beyond static head-to-head comparisons, this second experiment directly tests our hypothesis about the importance of data diversity. We investigate the dynamic interplay between different

---

[7]The predominantly 0-shot evaluation particularly benefits datasets like Nemotron 'diverse QA pairs' whose format directly aligns with many evaluation tasks.

data types to see how a model trained on one corpus adapts when another is introduced. This directly probes the synergistic potential discussed in our main paper. To do this, we took checkpoints of models pre-trained for 300B tokens on our MGA-Corpus and the Nemotron-CC (QA-slice), respectively. We then continued pre-training each model for an additional 30B tokens, mixing in data from the other corpus at a 1:1 ratio.

Table 14: Model Performance with Mixed-Corpus Continued Pre-training

| Experiment | Tokens | Wino. | C-QA | Hella. | MMLU | OBQA | PIQA | TriviaQA | ARC | AVG |
|---|---|---|---|---|---|---|---|---|---|---|
| mga | 300B | 52.4 | 31.5 | 42.4 | 31.3 | 38.0 | 68.6 | 4.2 | 45.3 | 37.3 |
| mga + mixct | 330B | 52.8 | 37.6 | 47.5 | 31.6 | 37.8 | 70.2 | 4.4 | 45.5 | 39.2 |
| *Change* | | *+0.4* | *+6.1* | *+5.1* | *+0.3* | *-0.2* | *+1.6* | *+0.2* | *+0.2* | *+1.9* |
| nemotron_qa | 300B | 52.6 | 38.8 | 49.0 | 31.4 | 38.8 | 70.8 | 9.2 | 47.0 | 40.7 |
| nemotron_qa + mixct | 330B | 51.8 | 37.8 | 47.8 | 31.8 | 36.6 | 70.6 | 3.5 | 45.9 | 39.1 |
| *Change* | | *-0.8* | *-1.1* | *-1.2* | *+0.4* | *-2.2* | *-0.2* | *-5.7* | *-1.1* | *-1.6* |

This cross-mixing experiment revealed two interesting and complementary phenomena:

**Synergy and Receptiveness of MGA.** When Nemotron-QA's structured data was mixed into the MGA-trained model, the model's average performance on downstream benchmarks significantly improved by 1.9 points. This suggests that the diverse, rich foundation built by MGA is highly effective and "receptive", readily absorbing the benefits of more specialized, format-aligned data.

**Distributional Path Dependence.** Conversely, when MGA's diverse data was mixed into the Nemotron-QA-trained model, its performance decreased by 1.6 points. This suggests that pre-training on a stylistically monolithic dataset can create a "path dependence", making it harder for the model to adapt when a different data distribution is introduced.

In conclusion, this experiment provides compelling evidence for our core argument. The strong performance of task-aligned data like Nemotron-QA is undeniable, but it does not represent a complete solution. Our findings show that the most effective approach is one that embraces variety. MGA's primary value lies in creating a robust, generalist foundation that is highly receptive to other data types. It helps build a model that is not locked into a single stylistic mode, effectively breaking down the data walls and paving the way for a more adaptable and capable AI. This synergy is the key to building next-generation base models.

## D.4 FURTHER ANALYSIS OF MODEL COLLAPSE

**Further Discussion of Section 4.3.3** For our analysis method in Figure 7, we define the token loss difference as $loss_{\text{diff}}^i = loss_{\text{synt}}^i - loss_{\text{real}}^i$, where i is the token index, synt/real is dataset used for model training. Note that we consistently use synthetic minus real, where a positive value indicates poorer prediction performance by the synthetic model on a given sample.

Since next token prediction is computed based on preceding context, we define the first anomaly position to identify where a model's prediction for tokens within the window begins to significantly deteriorate. The definition is as follows:

$$\text{first\_anomaly\_position} = \min\{p \mid \left| \frac{1}{w} \sum_{i=p}^{p+w-1} loss_{\text{diff}}^i \right| > |\mu| + k\sigma\},$$

where $w = \max(0.05 \times \text{seq\_length}, 1)$, $\mu = \text{mean}(loss_{\text{diff}}^i)$, $\sigma = std(loss_{\text{diff}}^i)$. Here, we employ the absolute value of the windowed average loss to identify significant performance degradation in either model. This approach enables the detection of notable prediction quality drops regardless of which model (synthetic or real) experiences the deterioration.

Finally, we define the normalized position, enabling fair comparisons across various sequence lengths:

$$\text{normalized\_position} = \begin{cases} \frac{\text{first\_anomaly\_position}}{\text{seq\_length}} \times 100\% & \text{if anomaly found} \\ -1 & \text{otherwise} \end{cases}$$

Below are example cases from English and Chinese documents. Figure 12 presents the token loss difference on each position. Example 2 and Example 3 show similar anomaly pattern, we can get the reason in Figure 13, that they are from the same website source contain identical boilerplate text about region selection and website localization at the end of their content.

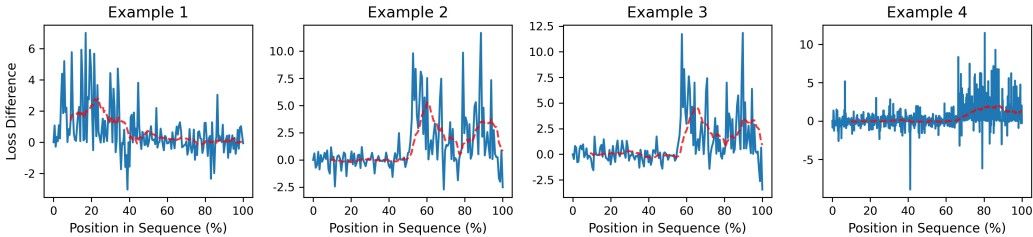

Figure 12: Random examples sampling from where mean$(loss^i_{\text{diff}}) > 0.5$, the synthetic-trained model fail to predict the tokens in later sequence positions.

This suggests potential noise in the data preprocessing pipeline, specifically in handling website navigation elements and localization prompts that should have been removed during content extraction.

While these examples demonstrate clear patterns of model behavior differences in handling noisy web data, we acknowledge that this analysis is limited to selected cases with apparent preprocessing artifacts. A more comprehensive evaluation across diverse data sources and quality levels would be necessary to fully understand the impact of synthetic training data on model performance.

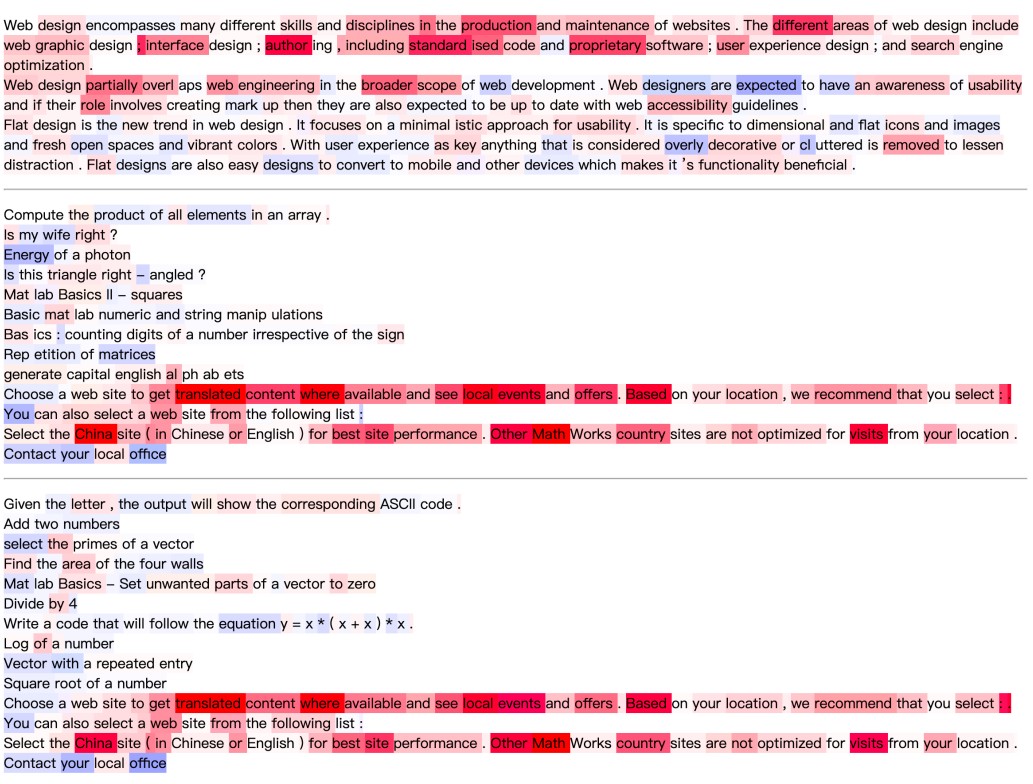

Figure 13: Corresponding cases sampled from Fineweb-Edu, which align with the loss patterns shown in Figure 12, with higher loss by synthetic-trained model highlighted in red.

英语 语法 极差，怎样 快速 学习？
办法 一：选《张 道 真 英语 语法》：1、从 句子 构造 着手，明白 句子 的 主要 成分，以及 不同 句子 成分 可由 什么 构造 但 当。2、研读 相应 构造 的 章节，有 大量 的 例句 参考。3、上 两步 学习 时 留神 谓语 动词 的 构造，参考 学习 相应 的 时态 和 语态。4、看 各种 句式 构造，熟练 称 述、疑问 等。5、看 复合 句 构造。6、看 非 谓语 动词 构造 和 其他 构造 构成 的 句子 与 复合 句 构造 的 关系。到 当前 自己 已经 摸到 了 窍门，学习 办法 自己 就会 获悉 了。办法 二：从《新 思想 英语 2》初步 自学，前 三个 单元 按 课程 要求 完结 后 主动 处理 了 语法 问题。日期 2个月 足够 了。此 办法 更 可取，除 语法 外 还 能够 结果 语感、表白、考试 做题 技巧、阅读、写作 等 方面。奇特 是 2个月 日期 就 能够 抵达 高考 110 分 的 程度。学习 英语 一定 要 踏实，一步 一个 脚印，任何 一●●而就 就能 掌握 英语 的 神话 那 都是 不 现实 的，尤其是 对于 基础 薄弱 的 初学者 而言。当然 了 如果 有 时间 者，我们 建议 最好 是 能 报 个 英语 培训班，因为 初学者 非常 需要 有 个 正确 的 引导 者，在 其 自学 的 过程 中 起到 推 波 助 澜 的 作用。关于 英语 培训机构 选 班 细节 的 问题，大家 可以 到 百度 上 搜 一下《废 哥 有科 测评》，是 最近 非常 火 的 英语 机构 测评 和 学习 规划 指导 专家，废 哥 做 了 八年 的 英语 机构 测评，体验 过上 百家 主流 英语 机构 的 课程，对于 每个 机构 的 课程、老师、收费 都 很 清楚，而且 能 给出 英语 学习 方面 的 很多 意见，不论是 报班 选课 都，还是 英语 学习 语感、语感 可以 让 我们 少 走 很多 弯路！学习 英语 时，不知道 如何 选课 报 班，现在 英语 班 五花八门，千万 不要 任意 而为，怎么 选课 防坑？哪些 班 实惠 有 效果？最近 很 火 的 这 篇 文章《成人 英语 培训 哪家 好？完整 分析 主流 班 优 劣势！》，上 百 个 英语 大牛 推荐，各大 主流 英语 班 的 情况 一览 无遗，参考 价值 非常 大！

---

开通 绿 钻 豪华 版 赠送 流量 包 还用 激活 么
不需要 激活 的，直接 充值 到 账 然后 可以 进行 使用 的。
流量 是 一个 数字 记录，记录 一台 手机 上 一个 网页 所 耗 的 字节 数，单位 有 B，KB，MB，GB。
怎么 计算，一个 英 文字 所 需要 1 B，而 一个 汉字 需要 2 B，一张 图片 一般 几 KB。用 手机 上 的 网页 一般 来说 是 几十 KB／每页，也就是 几万 B（1 KB＝1 0 2 4 B）。

---

车祸 引起 眼眶 骨折 做手术 还是 不 做？…
性别：男
年龄：10 岁
车祸 引起 眼眶 骨折 做手术 还是 不 做？怎么办 最后？
病情 分析：你 好，有些 小 范围 骨折，眼眶 内 陷 不 明显（不 超过 2 mm），外观 影响 不大；有的 骨折 范围 大，眼眶 内 陷 明显，且 合并 复视 等 症状，所以 是否 手术 要 视 骨折 范围 大小 而定。而复 视 问题，轻者 多 可 自行 恢复，不需 要 手术 治疗。
意见 建议：重度 眼眶 内 陷 影响 美容，复视 严重 会 影响 生活，此时 就 需要 手术 矫正 了。
孙 伟 主任医师
眼科 疾病 向 TA 提问
常见 的 眼科 疾病 有：中心 浆液 性 视网膜 病变、干眼症、夜盲 症、沙眼、白内障、糖尿病 视网膜 病变、结膜炎、老花 眼、虹膜 异色 症、视网膜 色素 变性、视网膜 中央 动脉 阻塞、视网膜 脱落、近视、远视、弱视、散光、针眼、盲症、青光眼、角膜 肿、青光眼、飞 蚊 症、上睑 下垂、肿 泡眼、角膜 病、疑难 眼 表 疾病、泪 道 疾病 等。不同 的 疾病 所 表现 的 症状 也 不太 相同，如 您 有 眼部 干涩、有 异物感、红肿、发痒、视物 模糊、畏 光、流泪 等 症状，最好 及 时 到 院 就诊 确认，以免 耽误 病情。

Figure 14: Chinese corpus samples with higher loss by synthetic-trained model in red.

# E PROMPTS AND CASES

## E.1 CASES

Table 15: Example outputs of SLM variants.

| Raw Document | | |
|---|---|---|
| In chemistry teaching, why is it said that skillfully capturing the optimal timing of ``movement-stillness'' transitions is key to success?
Because under the new curriculum philosophy, guiding students to adopt a learning method that combines ``movement and stillness'' transforms their thinking patterns and helps them learn through activities. Being able to skillfully capture the optimal timing for transitions between ``movement and stillness'' is key to success - seizing these perfect moments for transition leads to greater efficiency with less effort. | | |
| Genre-Audience Pairs | Tool Model | Reformulated Document |
| - Academic teaching paper, using rigorous academic style to deeply explore the importance, methods and practical case analysis of capturing the optimal timing of 'movement-stillness' transitions in chemistry teaching. Through references to relevant educational theories and actual teaching experiences, provide meaningful teaching insights for chemistry teachers.

- Middle school chemistry teachers, aged 25-45, predominantly female, with professional chemistry education background. They are patient and steady in character, have been teaching middle school chemistry for many years, and wish to improve teaching quality and help students better understand chemistry. | SLM-Base | **The Importance and Practice of Capturing Optimal Timing of "Movement-Stillness" Transitions in Middle School Chemistry Teaching**
**Abstract**: This paper aims to deeply explore the importance, methods, and practical case analysis of capturing optimal timing of "movement-stillness" transitions in middle school chemistry teaching from theoretical and practical perspectives. Through references to relevant educational theories and actual teaching experiences, it provides meaningful teaching insights for middle school chemistry teachers to help improve teaching quality and promote better student understanding of chemistry knowledge.
**Keywords**: Middle school chemistry teaching; "Movement-stillness" transitions; Optimal timing; Teaching methods
**I. Introduction**
With the continuous advancement of new curriculum concepts, middle school chemistry teaching is actively exploring more effective teaching methods to cultivate students' chemistry literacy and comprehensive abilities. Among these, having students adopt a learning method that combines "movement and stillness" has gradually gained attention, and skillfully capturing the optimal timing of "movement-stillness" transitions has become one of the keys to successful chemistry teaching.
**II. The Importance of Capturing Optimal Timing of "Movement-Stillness" Transitions**
(1) Alignment with Educational Theory
According to constructivist learning theory, student learning is a process of actively constructing knowledge through interaction with the external environment based on existing knowledge and experience. In middle school chemistry teaching, the learning method of "movement-stillness" transitions helps students concretize abstract chemical concepts through dynamic practical operations, leading to better understanding and mastery of chemistry knowledge. |
| | SLM-Strict | In the field of chemistry teaching, why is it said that skillfully capturing the optimal timing of "movement-stillness" transitions is key to success? Under the new curriculum philosophy, students are encouraged to use a learning method that combines "movement and stillness" to transform their thinking patterns and learn through activities. Being skillful at capturing the optimal timing of "movement-stillness" transitions is truly the key point of success - once this optimal timing is grasped, it can achieve twice the results with half the effort.
From a theoretical perspective, in middle school chemistry teaching, the concept of "movement-stillness" transitions aligns with students' cognitive patterns and helps stimulate their learning interest and initiative. Students can directly experience chemical phenomena through dynamic practical operations, deepening their understanding of knowledge; while in the static stages of thinking and summarizing, they can elevate perceptual knowledge to rational cognition and construct complete knowledge systems. In practice, teachers can analyze through specific cases. For example, when explaining the laboratory preparation of oxygen, teachers first guide students to observe the experimental apparatus and understand the function of each part - this is the "stillness" stage. Then, letting students personally conduct the experiment and observe phenomena such as bubble production and wood splint reignition is the "movement" stage. After students complete the experiment, organizing them to discuss problems encountered during the experiment and their solutions, summarizing and concluding, returns again to "stillness" reflection. Thus, through "movement-stillness" transitions, students not only master the knowledge and skills of oxygen preparation but also develop abilities in observation, analysis, and problem-solving. In conclusion, middle school chemistry teachers should fully recognize the importance of skillfully capturing optimal timing of "movement-stillness" transitions in teaching, and through reasonable design of teaching segments, integrate the learning method combining "movement and stillness" throughout the entire teaching process to improve teaching quality and help students better understand chemistry knowledge. |
| | SLM-Relaxed | In the field of education, for teachers of fundamental subjects, finding the key opportunities to guide students in effective learning is central to improving teaching effectiveness. Taking middle school chemistry teaching as an example, new educational concepts advocate students using flexible and diverse learning modes to transform their thinking pathways, enabling them to explore knowledge more independently in the learning process. Finding the perfect opportunity to guide students in achieving this thinking transformation during the learning process is undoubtedly key to successful teaching. Once this opportunity is successfully grasped, teaching effectiveness will be significantly improved, and students will be able to better understand and master the learned knowledge more deeply. |
| - Popular science story, using a lively and interesting narrative to tell story of how middle school students realizes the importance of capturing the optimal timing of 'movement-stillness' transitions in their learning process. Through specific interesting experiments, help students understand chemistry knowledge in a relaxed and enjoyable atmosphere, sparking interest in chemistry learning.

- Middle school students who find chemistry difficult, aged 12-15, both male and female, with below-average academic performance. They are generally lively and active in character, lack interest in chemistry but must study it, and hope to find ways to improve their chemistry grades. | SLM-Base | Once there was a middle school student named Xiao Ming, who was very lively and active, but initially had little interest in chemistry as a subject, and his grades were always below average.
One day in chemistry class, the teacher taught a particularly interesting lesson about a magical chemical change. The teacher said that during this change process, it was like having a mysterious little sprite at play - sometimes it was very quiet, and sometimes it became very active. This caught Xiao Ming's attention completely. He wondered when this little sprite would be quiet, and when it would become active.
Later, the teacher led everyone in doing an experiment. In the experiment, there were two small bottles - one bottle contained a transparent liquid, and the other bottle contained a magical powder. The teacher slowly poured the powder into the bottle containing the liquid. At first, nothing happened, and the liquid in the bottle remained perfectly still, as if it were sleeping. Xiao Ming and his classmates watched curiously, wondering when the little sprite would wake up.
Just when everyone was getting a bit impatient, suddenly, the liquid in the bottle started bubbling - it was as if the little sprite had suddenly woken up and started playing excitedly. This change came so suddenly that all the students were stunned, their eyes wide open.
The teacher smiled and told everyone that this was the "movement-stillness" transition. During this process, everyone had witnessed the sprite's change from quiet to active. The teacher also said that when studying chemistry, it's just like doing this experiment. You need to be good at capturing the optimal timing of "movement-stillness" transitions. If you can grasp it at that moment, you'll be able to better understand chemistry knowledge, and learning will be as effective with half the effort. |
| | SLM-Strict | Hey, classmates! Do you know that in the wonderful world of chemistry learning, there's something really important - that's being good at capturing the optimal timing of "movement-stillness" transitions, this is the key to success! Just like under the new curriculum concepts, teachers guide us to use a learning method that combines "movement and stillness," which can transform our way of thinking and help us learn through activities. In my chemistry learning process, I've deeply experienced this point.
Once, when we were doing an interesting experiment, the teacher told us to be good at catching the optimal timing of "movement-stillness" transitions. At first, I didn't quite understand. But later, I discovered that when certain phenomena in the experiment start to change, that's the optimal timing for "movement-stillness" transitions! As long as we catch this timing, we can achieve twice the results with half the effort in our experiments!
Hey classmates, don't you think it's amazing too? Let's discover more fun together in our chemistry learning journey! |
| | SLM-Relaxed | In a vibrant middle school classroom, there was a group of lively and active students, aged between 12-15, both boys and girls, with below-average academic performance. Among them was a student named Xiao Le, who had no interest in chemistry but had to study it, and was always looking for ways to improve his grades.
One day, in the school laboratory, Xiao Le watched the teacher perform a magical experiment. The teacher first put some ordinary-looking powder into a transparent bottle, then added a liquid. Immediately, it was like a lively party started in the bottle, with lots of colorful bubbles appearing - this completely captivated Xiao Le and his classmates.
Xiao Le thought to himself that this was just like how he played in the playground - sometimes resting quietly, sometimes running around wildly - isn't this a combination of "movement and stillness"? Moreover, he discovered that during this process, the moment when the powder and liquid made contact was the most magical moment, like finding a hidden treasure.
From then on, every time Xiao Le did chemistry experiments, he would pay special attention to this optimal timing of "movement-stillness" transitions. He found that when he caught this timing, he could understand chemistry knowledge better. His chemistry grades gradually improved, and his interest in studying chemistry grew stronger and stronger. |
| - Parenting education guide, using easy-to-understand language style combined with real-life scenarios to explain the concept of capturing optimal timing of 'movement-stillness' transitions in chemistry teaching. Provide simple parent-child interaction methods to help parents guide their children in understanding and applying this learning method in daily life, while improving parent-child relationships.

- Parents, aged 30-50, both male and female, with diverse occupations. They care about their children's learning and growth, hope to understand what their children are learning at school to better assist with their studies, but have limited chemistry knowledge themselves. | SLM-Base | Dear parents, do you know that in your child's chemistry learning process, there's an important method called the "movement-stillness combined" learning approach? It's just like in our daily life - sometimes children need to quietly think about problems, and other times they need to engage in hands-on experiments or practical activities. So why is skillfully capturing the optimal timing of "movement-stillness" transitions key to success?
Think about it - at school, teachers use this learning method to transform children's thinking patterns, helping them learn through activities. For example, when students are learning about chemical elements, they might first quietly listen to the teacher explain the properties and characteristics of elements - this is the "stillness" phase. Then, the teacher will arrange experiments for children to perform hands-on operations and observe chemical phenomena - this enters the "movement" phase. If parents can help their children capture the optimal timing of these "movement-stillness" transitions, their chemistry learning will become twice as effective with half the effort!
So how can we help children achieve this in daily life? It's actually quite simple. For instance, after your child learns a chemistry concept, you can ask: "Sweetie, can you think of any phenomena in daily life that relate to this concept?" This is a small interaction that guides children from "stillness" to "movement." When your child starts thinking and answering, you can encourage them to do some simple experiments to verify their ideas, like conducting a small experiment using materials at home. This way, children can better understand and apply the "movement-stillness" learning method in their daily lives, and during this process, your parent-child relationship will grow closer too! |
| | SLM-Strict | As for why skillfully grasping the optimal timing of "movement-stillness" transitions is key to your child's success in learning chemistry - here's something to consider. The new curriculum approach encourages children to use a learning method that combines "movement and stillness," as this can transform their thinking patterns and help them learn through activities. Being able to capture these transition moments is crucial - when you catch these perfect moments, your child's learning becomes twice as effective with half the effort.
So how can we parents help our children understand and apply this learning method in daily life? Let me share some simple parent-child interaction methods. For example, we can do a small experiment together with our children, like observing the reaction between baking soda and white vinegar. During this process, the hands-on operation is the "movement" part, while observing the phenomena and thinking about the reasons is the "stillness" part - see how this combines "movement and stillness"? You can guide your child to pay attention to these processes, and gradually, they'll better master this learning method. Plus, during this process, your parent-child relationship will naturally grow stronger! |
| | SLM-Relaxed | "A Parent's Guide to Supporting Children's Learning and Growth"
During a child's learning and growth process, there is an important learning method called the movement-stillness transition method. It's like a magical key that can help children better understand and master knowledge, enhancing their learning effectiveness. So, how do we guide children to master this method?
For parents, although their professional knowledge may be limited, they can help children understand through various scenarios in daily life. For example, during a family trip, when children encounter different scenery and activities, some children might initially run and play excitedly - this is the "movement" state. Then when they see beautiful scenery or encounter interesting things, they stop to carefully observe and think - this enters the "stillness" state. Parents should be good at recognizing these transitions between states in their children, guiding them at appropriate times to help them understand that this process of "movement-stillness" transition is actually the learning process. Through such parent-child interactions, parents can not only better understand what their children are learning at school but also improve their parent-child relationship, accompanying their children in healthy and happy growth. |

## E.2 PROMPTS

Although the term "rewrite" is used in some prompt templates as the editing instruction, it serves the same function as "reformulate" discussed in sections above, which aims to maintain the core meaning of the documents while only optimizing its expression.

| |
|---|
| # strict version
You are a text polishing expert. You will polish text based on the given [Genre] and [Audience].

When polishing, you must follow these 4 rules:
1. Read through the entire text and polish it according to the requirements of the given [Genre] and [Audience]
2. The degree of polishing should not be too heavy – just aim to satisfy the requirements of [Genre] and [Audience] as much as possible
3. Double–check that the polished text is suitable for the audience described in [Audience]!
4. Pay attention to the frequency of modal particles – the text should not contain too many modal particles |
| # relaxed version
You are a creative expert skilled at transforming materials into creative inspiration and building independent, complete, and highly original texts.

Requirements:
1. Read through the original text thoroughly, extract several key themes/keywords, transform to abstract or universal concept inspiration, then generate entirely new text constructions.
2. Extract content from [Audience] and [Genre] sections, but don't be constrained by them directly, just use them as creative inspiration.
3. Create and reformulated text around points 1/2, and build new meaning from details to the whole structure. |

Figure 15: two different prompt templates, we keep the input aligned with MGA strategy, using raw text, genre, audience to fill the template.

```
#############
#Identity and Capabilities#
You are a content creation expert, specializing in text analysis and rewriting, capable of adapting content
based on varying ''genres'' and ''audiences'' to produce ''diverse'' and ''high–quality'' texts. Your English
writing is at native editor level, and you will output your rewritten texts in English. International audiences
particularly enjoy your work, which receives widespread readership and circulation, earning unanimous
acclaim from the industry for your capabilities!

#############
#Workflow#
Please utilize your analytical and writing abilities to rewrite the text based on the original content and given ''
genre'' and ''audience''. Before beginning the rewrite, you will consider the following requirements:

1. First, read through the original text thoroughly, identify its information content and value, and consider
how to prevent any loss of information points and value in the rewritten text
2. Focus on the original content, combine it with the given ''genre'' requirements, and rewrite the text
following the descriptions, content modules, language requirements, and other stylistic elements specified in
the ''genre'', to form an initial draft
3. Polish the initial draft according to the given ''audience'' requirements, and generate the final rewritten text
in English
4. Refine the rewritten text to match native English speakers' reading habits and expression patterns

#############
#Detailed Requirements#
Please ensure you follow the three workflow requirements above, then generate the final English rewritten text
according to these detailed requirements.
The given ''audience'' is <<<{audience}>>>.
The given ''genre'' is <<<{genre}>>>.

#############
#Raw Text#
{raw_text}
```

Prompt 1: reformulation prompt template.

#############
#Identity and Capabilities#
You are a content creation expert, specializing in text analysis and rewriting, skilled at adapting content based on varying [genres] and [audiences] to produce ''diverse'' and ''high−quality'' texts. Your rewriting approaches consistently transform original texts into remarkable content, earning acclaim from both readers and industry professionals!

#############
#Workflow#
Please utilize your imagination and creativity to generate 5 pairs of [genre] and [audience] combinations suitable for the original text. Your analysis should follow these requirements:

1. First, analyze the characteristics of the source text, including writing style, information content, and value
2. Then, consider how to preserve the primary content and information while exploring possibilities for '' broader audience engagement'' and ''alternative genres''

#############
#Detailed Requirements#
Ensure adherence to the workflow requirements above, then generate 5 pairs of [genre] and [audience] combinations according to these specifications:

Your provided [genres] should meet the following requirements:
1. Clear Genre Definition: Demonstrate strong diversity; include genres you've encountered, read, or can envision
2. Detailed Genre Description: Provide 2−3 sentences describing each genre, considering but not limited to type, style, emotional tone, form, conflict, rhythm, and atmosphere. Emphasize diversity to guide knowledge adaptation for specific audiences, facilitating comprehension across different backgrounds. Note: Exclude visual formats (picture books, comics, videos); use text−only genres.

Your provided [audiences] should meet the following requirements:
1. Clear Audience Definition: Demonstrate strong diversity; include both interested and uninterested parties, those who like and dislike the content, overcoming bias toward positive audiences only
2. Detailed Audience Description: Provide 2 sentences describing each audience, including but not limited to age, occupation, gender, personality, appearance, educational background, life stage, motivations and goals, interests, and cognitive level

#############
#Response#
```
{
    ''audience_1'': audience1,
    ''genre_1'': genre1,
    ''audience_2'': audience2,
    ''genre_2'': genre2,
    ''audience_3'': audience3,
    ''genre_3'': genre3,
    ''audience_4'': audience4,
    ''genre_4'': genre4,
    ''audience_5'': audience5,
    ''genre_5'': genre5
}
```

#############
#Input#
{raw_text}

Prompt 2: genre-audience pairs prompt template.

#############
#Identity and Capabilities#
You are a Content Reviewer, skilled at analyzing texts and keenly identifying and analyzing the relationships, similarities, and differences between two texts. Your thorough analysis of each pair of texts, with attention to every detail, provides great convenience for subsequent review work!

#############
#Thinking Process#
Please fully utilize your analytical abilities, review capabilities, and deep thinking skills to analyze the ''Rewritten Text'' against the ''Original Text'' as a benchmark, ultimately providing analysis and scoring for [A]. You will follow these steps for detailed consideration:

1. First, you will read through the original text thoroughly, identifying the information points in the ''Original Text''
2. You will also read through the rewritten text thoroughly, identifying the information points in the ''Rewritten Text''
3. Compare the information in both texts' content. The ''Rewritten Text'' is allowed to have new information points, different writing styles, expression styles, order, and focus from the ''Original Text''. As long as it is created based on some information points from the ''Original Text'', it is considered good for [A]
4. After careful analysis and review, please clearly list the connections and differences between the two texts, and based on this, provide final analysis and scoring for [A]

#############
#Detailed Requirements#
The scoring judgment for [A] must follow these standards:
1. The ''scoring range'' is 1–5 points. You need to analyze and grasp each aspect mentioned in #Thinking Process#, and differentiate scores accordingly. Be strict, don't be too lenient with scoring!
2. The ''Rewritten Text'' is allowed to differ from the ''Original Text'' in writing style, expression style, and focus! This cannot be a basis for deducting points!
3. The ''Rewritten Text'' is allowed to omit some information from the ''Original Text''! It is not required that all information from the ''Original Text'' appears in the ''Rewritten Text''! This also cannot be a basis for deducting points! If this is the only issue, please give a full score of 5 points.

In scoring [A], the following situations **NOT reduce** the score for [A]:
1. The ''Rewritten Text'' can include information points not present in the ''Original Text''
2. The added content in the ''Rewritten Text'' significantly deviates from the core information of the ''Original Text''
3. The expression style, order, and focus of the ''Rewritten Text'' differ from the ''Original Text''

In scoring [A], the following situations **WILL reduce** the score for [A]:
1. The information points in the ''Rewritten Text'' differ so greatly from the ''Original Text'' that it's not recognizable as being rewritten from the ''Original Text''
2. The ''Rewritten Text'' contains none of the information points from the ''Original Text''

#############
#Original Text#
{raw_text}

#Rewritten Text#
{rewritten_text}

#############
#Response Format#
{
    ''A'':{
        ''analysis'': ''xxx'', provide reasons for point deductions
        ''score'': 1, 2, 3, 4, or 5
    },
}
#############

Prompt 3: Full LLM judger prompt.

## F    USE OF LARGE LANGUAGE MODELS

During the preparation of this manuscript, we utilized a large language model (LLM) as a writing assistant. The LLM's role was primarily focused on improving the clarity, precision, and readability of the text. This included tasks such as correcting grammar and spelling, refining sentence structure for better flow, and suggesting alternative phrasing to enhance the academic tone.

The core scientific contributions—including the initial research ideation, the design of the MGA framework, the experimental methodology, and the interpretation of results—were conceived and executed entirely by the human authors. The LLM did not contribute to the research ideas or the analysis presented. The authors have meticulously reviewed, edited, and validated all LLM-assisted text to ensure its scientific accuracy and take full responsibility for the final content of this work.

