# OpenReview forum: "Reformulation for Pretraining Data Augmentation"
_ICLR.cc/2026/Conference — ICLR 2026 Poster_

### Official Review · Reviewer_eWsn · 2025-10-29

**Soundness:** 3
**Presentation:** 3
**Contribution:** 3
**Rating:** 6
**Confidence:** 3

**Summary:**

This paper introduces Massive Genre Audience (MGA), a reformulation method to augment training corpora based on generating multiple diverse genre-audience pairs per inference pass through a prompting-based method. This is done with the intention of mitigating the influence of data repetition during training that often leads to performance degradation particularly in data scarcity scenarios. The authors proceed to analyze the effectiveness of their method (including whether the diversity introduced is actually beneficial) and how it synergizes with alternate synthetic data strategies. Models at different scales are evaluated across a wide range of datasets.

**Strengths:**

1. The paper is well organized and clear. Specific research questions are laid out and experiments are directly designed to support each one in a clear manner.
2. Comprehensive results across many datasets and models at 5 different scales.
3. The paper emphasizes and shows the synergistic nature of their method.
4. Authors state they will release MGACorpus and all artifacts for reproducibility.
5. The motivation of addressing negative influences of data repetition during training is important.

**Weaknesses:**

1. The method is based on prompt engineering and as such there are some potential issues with bias introduced by using a solely LLM based system. For example, when the LLM is used for judging, what sorts of bias is being introduced? Was some form of human evaluation done or some alternative test to determine whether the scores produced by the LLM align with the actual quality of the text. Furthermore there should be some evidence that actual diverse outputs are being generated by the method. I believe that there should also be some results on text heterogeneity quantification to pinpoint the actual heterogeneity introduced by the sampling method which would allow us to concretely correlate with the performance.

2. Tied to the above, some discussion on specific design decisions is warranted, namely the choices behind the genre audience pairs. Also it is stated that the SLM "executes a “one-pass-for-many” strategy ... since repeated sampling requests to a model can yield highly similar outputs" but doing so can lead to bias since later generated pairs are conditioned on the early ones compared to random sampling (although as mentioned some means to stop potential redundancy is needed). If heterogeneity could be quantified and these two methods compared this assertion could be validated.

3. Given the similarities in the idea, some discussion of previous works on synthetic data creation using personas [1] is warranted in the paper.

4. Only one model family is tested (SmolLM) for MGA-Expansion which presents a limitation in determining how well the method will work on alternative architectures.

5. Table 3 is not referred or discussed anywhere in the text.

[1] Tao Ge et al. Scaling Synthetic Data Creation with 1,000,000,000 Personas. Arxiv. 2024

**Questions:**

See Weaknesses

---

> ### Author Response · Authors · 2025-11-14
>
> **Dear Reviewer eWsn**,
>
> Thank you for your constructive and detailed feedback. We appreciate your positive assessment of the paper's organization, comprehensive results, and importance.
>
> You raised crucial points regarding methodological validation, which we address below with new analyses.
>
> ---
>
> ### **W1: Bias, Human Evaluation, and Heterogeneity Quantification**
>
> This is a critical point. We agree rigorous validation is needed and provide both qualitative and quantitative analyses.
>
> **1. Qualitative: Human-in-the-Loop Validation**
> To validate our LLM-based judging, we conducted a multi-round human-in-the-loop review.
> We randomly sampled 30 source documents and their 150 corresponding MGA rewrites.
> Our human reviewers tuned the prompt template until the LLM judge's scores and its *reasons* had **over a 90% agreement rate** with human judgment, confirming our scoring standard aligns with human intuition.
>
> **2. Quantitative: Heterogeneity Quantification**
> To move beyond claims and *quantify* diversity, we introduce two metrics:
> * **Reference Similarity:** The average cosine similarity between each synthetic document and its original source.
> `RefSim = avg(sim(docs_ori, docs_syn))`
>
>
> * **Heterogeneity Score:** Measures the internal diversity of corpus. It is defined as $ S = 1 - \left( \frac{1}{|D|*(|D|-1)}\sum_{i \neq j}\text{sim}(\text{doc}_i, \text{doc}_j)\right) $. **A higher score means more diversity.**
>
> Experiments on this trade-off confirmed our design choice:
>
> | Setting (Prompt Strategy)   | Reference Similarity (vs. Original) | Heterogeneity Score (Internal Diversity) |
> |:--- |:------------------------------------|:-----------------------------------------|
> | Original Data| 1.0   | 0.3355|
> | **Base (Our Method)** | **0.8979 (Balanced)**| **0.3390 (Highest)** |
> | Strict (Enforce high similarity) | 0.9392 (Highest)| 0.3326 (Diversity < Original)  |
> | Relaxed (LLM 'free' sampling) | 0.8319 (Lowest) | 0.3063 (Lowest) |
>
> **Conclusion:** This data reveals a crucial trade-off.
> * The "Strict" policy is faithful but sacrifices diversity (0.3326).
> * The "Relaxed" policy, as you hypothesized, suffers from severe homogenization (mode collapse) (0.3063).
> * Our "Base" MGA strategy finds the sweet spot, maximizing heterogeneity (0.3390) while maintaining source relevance (0.8979).
>
> ---
>
>
> ### **W2: Quantify "One-pass-for-many"**
>
> You raise an excellent question about whether our "one-pass-for-many" strategy introduces conditioning bias compared to repeated single sampling ("one-pass-for-one"). We ran a direct comparison on 1k validation examples and found our method *significantly increases* diversity:
>
> | Metric | **One-pass-for-many (Ours)** | **One-pass-for-one** |
> |:--- |:--- |:--- |
> | Unique Genres  | **913**| 779  |
> | Unique Audiences| **871**| 715  |
> | Unique Pairs   | **987**| 951  |
> | Type-Token Ratio (TTR) | **0.1213**   | 0.0839|
> | TFIDF Dissimilarity    | **0.9751**   | 0.9572|
>
>
> This validates our design. Our approach acts as a diversity catalyst, generating more unique pairs and higher lexical (TTR) and semantic (TFIDF) diversity.
>
> ---
>
> ### **W3: Discussion of Ge et al. [1] (Personas)**
> Thank you for this excellent suggestion. We apologize for the oversight during document compression for the submission.
> The idea of using personas to enhance diversity was indeed an important inspiration from Ge et al. [1].
> We will cite and discuss it, clarifying the key distinction: **Ge et al. [1] focused on generating Q&A pairs, whereas MGA adapts the concept for the pre-training domain by reformulating existing web text**.
>
> ---
>
> ### **W4: Generalization to Other Architectures**
>
>
> This is a fair point. We confirmed these findings on an internal MoE model family.
> While we cannot disclose architectural details due to institutional policy, we can share the key results,
> a consistent, positive impact from MGA:
>
> | Experiments | Model Size (Training Tokens)  | **Average Score** | **Difference** |
> |:--- |:--- |:--- |:--- |
> | Baseline| 6.8B-A680m  (500B)  | 45.31||
> | **+ MGA**    | 6.8B-A680m (500B)   | **46.99** | **+1.69**|
> | Baseline| 12.5B-A1.25B (500B)  | 51.97||
> | **+ MGA**    | 12.5B -A1.25B (500B) | **53.78** | **+1.81**|
>
>
>
> This demonstrates that MGA's benefits are not unique to the SmolLM (Dense) architecture and generalize effectively to MoE models.
>
> ---
>
> ### **W5: Table 3 Not Discussed**
>
> You are correct; this was an oversight. Table 3 corresponds to the "SLM variant" discussion (line 378), showing how Prompt choices impact the distribution of rewrites. We will add a direct reference to clarify this.
>
> ---
>
> We hope these new results and clarifications thoroughly address your concerns and demonstrate the robustness and novelty of MGA. We are grateful for the feedback, which has significantly strengthened our paper.
>
> [1] Tao Ge et al. Scaling Synthetic Data Creation with 1,000,000,000 Personas. 2024.

---

> ### Author Response · Authors · 2025-11-27
>
> **Dear Reviewer eWsn,**
>
> Thank you again for your constructive suggestions regarding bias and diversity quantification.
>
> We wanted to gently check if you had a chance to review our response, particularly the new quantitative analyses you requested.
>
> We believe these additional results provide the solid evidence needed to address your concerns about validity and generalization. We look forward to hearing your thoughts.
>
> Best regards, The Authors

---

### Official Review · Reviewer_HLbn · 2025-10-31

**Soundness:** 4
**Presentation:** 4
**Contribution:** 3
**Rating:** 8
**Confidence:** 3

**Summary:**

This paper proposes a new way of synthetically augmenting pretraining data through Genre Audience reformulation.
This paper works, and its numbers are solid. The science is also thorough.

**Strengths:**

1. The greatest strength is the scaling plots, which demonstrate consistent performance improvement in the token-matched regime. Really nice! Figure 5 was particularly great. The training noise seems relatively less which is good for trusting the downstream evals.
2. Really nice overall benchmark numbers. Grounds the work with good numbers to help set this up relative to other methods.  Achieving a MMLU score of 40.7 is great for the 1.7B model class.
3. Beating finding more hq data is really good result.  Particularly Figure 3 is strong! The graphs showed consistent improvement across all scales, beating the use of a more original dataset.
4. Also, a good set of benchmark coverage. LightEval and LM-Harness are both great tools for eval and using the set of evals like MMLU is great.

**Weaknesses:**

My main gripe is that I would have liked to see a few more baselines of synthetic data/rephrasing. WRAP, for example, would have been nice to show. I understand that this would have been expensive at the higher token counts, but at the lower token counts, it would have been nice. There are great synthetic pretraining data techniques. Could you add WRAP or just adding other synthetic post-training datasets (nemotron for example) would be great?

**Questions:**

N/A

---

> ### Author Response · Authors · 2025-11-14
>
> **Dear Reviewer HLbn**,
>
> We are thrilled by your strong endorsement and your appreciation of our scaling properties and training analysis. We share your excitement about the potential of synthetic data and are grateful for your recognition of our rigorous evaluation.
>
> Below, we address your suggestion regarding additional baselines.
>
> ---
>
> ### **W1: Comparison with WRAP and Post-training Datasets**
>
> We fully agree that comparing against a broader range of synthetic strategies is valuable. We address this in three aspects:
>
> #### **1. Nemotron-Syn (V1) as a Scaled-Up WRAP Proxy**
> We prioritized **Nemotron-Syn (V1)** as our primary baseline because it effectively represents the state-of-the-art evolution of the **WRAP** methodology (wiki and qa style) at scale.
> Comparing MGA against the breakdown of Nemotron's sub-sources (see Appendix D.3.1) reveals:
> * **Comparison with Nemotron:** MGA outperforms 5 out of the 6 sub-sources in Nemotron-Syn (V1).
> * **Comparison with Cosmopedia:** MGA outperforms Cosmopedia under the same token budget.
> * **The "QA" Exception:** The only subset MGA does not beat is the QA-style rephrasing dataset, which, as noted in our response to Reviewer W91b, dominates the aggregated performance due to format alignment.
>
> #### **2. Reproducibility and Standardized Baselines**
> Your suggestion highlights a broader challenge in the field: **The lack of a standardized testbed for Synthetic Data.**
>
> Unlike real data research (e.g., DCLM), synthetic data research often suffers from "apples-to-oranges" comparisons regarding Source Domain, Target Format, and Evaluation Metrics. Furthermore, meaningful research in this area is compute-intensive, often restricting it to frontier labs.
>
>
> Currently, Nemotron-V2-Syn (3T) and MGA (770B) are the only two massive-scale synthetic pre-training corpora.
> But Nemotron-V2 doesn't provide a full meta info about the source text and target task for synthesis process.
> By releasing the second-largest synthetic corpus and our generation pipeline, we aim to provide a necessary public baseline for the broader community.
>
> #### **3. On Integrating Post-Training (Instruction) Data into Pre-training**
> Regarding your suggestion to include post-training datasets (like Nemotron instructions), this touches upon a rapidly evolving research frontier: **Instruction Pre-training**.
>
> * To our knowledge, Physics of Language Models [1] was the first to demonstrate the necessity of synthetic instruction data for knowledge extraction.
> * More recent works [2, 3] have begun to mix SFT data into pre-training. Notably, Nemotron V2 shifts the SFT ratio from 3.1% to 42.9% in pre-training stages, while did not report the ablations.
>
> While promising, determining the optimal mixing ratios and dynamic schedules for instruction data is an open research question with many uncontrolled variables.
> Therefore, we focused this paper on the pure pre-training signal of MGA. However, we agree this is the logical next step and aim to explore in future work.
>
>
>
> Thank you again for recognizing the value of our work!
>
> ---
>
> [1] Physics of language models: Part 3.1, knowledge storage and extraction. 2023.
> [2] Instruction pre-training: Language models are supervised multitask learners. 2024.
> [3] NVIDIA Nemotron Nano 2: An Accurate and Efficient Hybrid Mamba-Transformer Reasoning Model. 2025.

---

> ### Author Response · Authors · 2025-11-27
>
> **Dear Reviewer HLbn,**
>
> Thank you again for your strong support and appreciation of our scaling analysis.
>
> We just wanted to ensure that our response regarding the additional baselines (comparison with WRAP and Nemotron breakdown) and the discussion on instruction pre-training adequately addressed your comments.
>
> We hope our clarifications further solidify your positive assessment of our work.
>
> Best regards, The Authors

---

### Official Review · Reviewer_W91b · 2025-11-01

**Soundness:** 3
**Presentation:** 3
**Contribution:** 2
**Rating:** 6
**Confidence:** 3

**Summary:**

This paper introduces a synthetic data generation pipeline for pretraining language models. Their pipeline, MGA (Massive-Genre Audience), asks a lightweight 3.3B MoE model to rewrite source text for different “genres” (e.g. story, dialogue, textbook) and audiences “girl”, “grandpa”, and “teacher”. MGA provides better scaling properties for training language models than training on the original data alone.

**Strengths:**

Most of the experiments in this paper are done very thoroughly, and results are presented in a very neat and orderly manner. I appreciate that the authors release an open-source dataset and also release tooling and other artifacts that make this work reproducible.

**Weaknesses:**

A key thing I wanted to know in this paper was whether the author’s proposed approach, MGA, outperforms alternative approaches for generating synthetic data. Even though the authors say that “MGA is not in competition with but is complementary to other synthetic data methodologies” (line 370), the authors should properly demonstrate MGA’s utility in the current landscape of synthetic data generation strategies. I am not sure if I got a clear answer to this question from the current version of the paper.

Section 4.3.1. Claims that MGA can complement existing synthetic data approaches. There, a plot shows the following, as I quote:
> - Baseline: A high-quality real dataset fineweb-edu.
> - Exp A: 35% token budget replaced by Nemotron-CC-HQ synthetic corpus (+Nemotron-Syn).
> - Exp B: 35% token budget replaced by MGACorpus (+MGA).
> - Exp C: 70% token budget replaced by an equal combination of Nemotron-Syn and MGACorpus data (+Nemotron-Syn +MGA).

The authors’ results show that though A outperforms B, C outperforms A and C outperforms B. However, one key experiment missing here is whether a 70% token budget replaced by only Nemotron-Syn outperforms C, because it if it does, then it makes the inclusion of any MGACorpus/MGA data not necessary and potentially detrimental.

**Questions:**

The authors have a nice analytical discussion section around data reformulation diversity, learning characteristics, and mode collapse – it does make me curious if some of the conceptual findings here generalize to other synthetic data generation processes.

I may have accidentally missed this somewhere, but from where did you source the range of possible “genres” (structural/stylistic formats, e.g. dialogue) and “audiences” (reader profiles, e.g. teacher) for your pipeline?

---

> ### Author Response · Authors · 2025-11-14
>
> **Dear Reviewer W91b**,
>
> Thank you for your insightful review. You raise a critical question: *Does a 70% Nemotron-Syn budget (Hypothetical Exp A-70%) outperform the mixed strategy (Exp C), potentially rendering MGA unnecessary?*
>
> ### **W1: Comparison with Nemotron-Syn & Necessity of MGA**
>
> We address this using an equivalent proxy comparison at the 35% budget, which were finalized after the submission deadline.
> However, before presenting the results,we must point out that  **a direct performance comparison between Nemotron-Syn and MGA is confounded for two reasons**:
>
> 1. **The "Source Data" Advantage**: Nemotron-Syn is derived from Nemotron-CC-HQ, while MGA is derived from FinewebEDU.
>    1. Model trained on CC-HQ outperform that trained on FinewebEDU by a **massive margin** (MMLU `+16.1` on 8B models) [2].
>    2. In Appendix D.3.1, we show for the same target WRAP-medium, data derived from HQ source outperforms LQ source by `4.91`.
> 2. **The "QA Format" Advantage**: QA-style rephrasing is a dominant component of WRAP [1] and Nemotron-Syn [2]. As demonstrated in [3], transforming data into QA formats significantly alters how models encode knowledge, making it more retrievable.
> Crucially, this format directly aligns with downstream benchmarks, creating a format-level advantage.
>
> Despite operating with a massive "source quality deficit", Exp C.2 is neck-and-neck with Exp A (`46.01` vs. `46.14`):
>
> |   | Data Composition (35% Budget) | Avg. Score |
> | :--- | :--- | :--- |
> | Exp A | 35% Nemotron (Full) | 46.14 |
> | Exp C.2 | 17.5% Nemotron + 17.5% MGA | 46.01 |
>
> When we remove the QA sub set (Exp B vs. Exp A.2), the two methods perform equivalently (`45.46` vs `45.52`):
>
> |   | Data Composition (35% Budget) | Avg. Score |
> | :--- | :--- | :--- |
> | Exp B | 35% MGA | 45.46 |
> | Exp A.2 | 35% Nemotron (**w/o QA**) | 45.52 |
>
> This proves MGA's robustness via reformulation alone rather than hyper-curated sources.
> Ultimately, considering the finite supply of the expensive unique synthetic tokens (Nemotron-CC-HQ: 500B vs. MGA: 770B), combining strategies (Exp C) is essential to provide more data while mitigating single-method bias and maximize data diversity.
>
> ---
>
> ### **Q1: Generalizing conceptual findings**
>
> This is a deep and valuable question. While the hypothesis that synthetic data naturally aligns with Next-Token Prediction (NTP) objectives has been posited in recent works like Phi-4 [4] and Beyond Web [5], these studies primarily discussed the concept without extensive experimental isolation or granular case analysis.
>
> Moreover, our case studies found **no evidence linking these distinct patterns solely to MGA's specific architecture**.
> Training on **MGA even aids val loss of Cosmopeida, which is generated under a total different pipeline** [6].
> This suggests that such learning dynamics might be a generalizable feature of synthetic data distinct from natural text, consistent with the hypothesis in [4, 5], rather than a byproduct of our specific method.
>
> Regarding mode collapse, we honestly acknowledge this is *not fully understood*, especially in the new paradigm of real-world data generation. Much of the new text on the internet is LLM-generated but has undergone *human rejection sampling* (i.e., users carefully editing prompts until they get a satisfactory result). **Our work aims to be a first step in considering the *prompt's role* (via genre/audience) in studying diversity and mode collapse**, reflecting this new human-AI collaborative creation process. We will clarify this limitation and context in the discussion.
>
> ---
>
> ### **Q2: Source of "genres" and "audiences"**
>
> Thank you for prompting this clarification. We apologize if this was not clear. As detailed in the Methods section, a key feature of MGA is that **we do not use a fixed seed bank or manually curated list of genres/audiences.**
>
> The strategy is *adaptive*, the GA-SLM generates these pairs on the fly based on the source text. This is a crucial design choice that makes our pipeline **lightweight and easily scalable**, as it removes the need to collect, manage, and maintain a complex and potentially biased seed system. We will emphasize this point more clearly in the final version.
>
> We hope these clarifications and new experimental results resolve the concerns and underscore MGA's contribution to the field of synthetic pretraining data.
>
> ---
>
> [1] Rephrasing the web: A recipe for compute and data-efficient language modeling. 2024.
> [2] Nemotron-cc: Transforming common crawl into a refined long-horizon pretraining dataset. 2025.
> [3] Physics of language models: Part 3.1, knowledge storage and extraction. 2023.
> [4] Phi-4 technical report. 2024.
> [5] Beyondweb: Lessons from scaling synthetic data for trillion-scale pretraining. 2025.
> [6] Cosmopedia: how to create large-scale synthetic data for pre-training. 2024.

---

> > ### Author Response · Authors · 2025-11-27
> >
> > **Dear Reviewer W91b,**
> >
> > As the discussion phase is progressing, we wanted to respectfully follow up to ensure that our response and new experimental results regarding your key concern have been clear.
> >
> > We would greatly appreciate it if you could let us know if these new results resolve your concerns. We are happy to engage in further discussion if you have any remaining questions.
> >
> > Best regards, The Authors

---

### Author Response · Authors · 2025-12-01
**Summary of Contributions and Rebuttal Updates**

**Dear Area Chair and Reviewers,**

We thank all reviewers for their constructive feedback and their recognition of our work’s "thorough scaling analysis," "solid numbers," and "excellent presentation."
Given the unique situation, we provide this summary to highlight our core contributions and how the rebuttal has further validated our claims, particularly regarding state-of-the-art comparisons and empirical insights into synthetic data.

### **Core Contributions & Strengths.**
This paper investigates a reformulation-based strategy to navigate the challenges of data scarcity and the 'Data Wall'.
Our method, MGA, reformulates web text to enable continued scaling while mitigating the degradation caused by data repetition.

- **Validated Scaling Properties.** Reviewers highlighted our "thorough" and "excellent" scaling analysis (Reviewer HLbn, W91b), proving MGA consistently outperforms data repetition and upsampling across model sizes (up to 13B parameters and 1T tokens).
- **High-Performance Baseline.** We achieve strong results (e.g., 40.7 MMLU on a 1.7B model), which Reviewer HLbn noted as "excellent for the model class."
- **New Insight on Mode Collapse.** We offer a critical insight into why synthetic data fails. Our analysis of "Relaxed" (free sampling) vs. "MGA" (balanced) strategies reveals that mode collapse is not solely a model limitation, but a prompt engineering failure. By systematically constraining the Prompt, we prove that pipeline design is the primary lever to prevent homogenization.
- **Major Community Resource.** We will provide the first fully reproducible, massive-scale synthetic data, releasing the 770B token MGACorpus and key artifacts. This addresses the lack of public baselines in a field dominated by proprietary datasets.


### **Rebuttal Highlight I: Superiority over SOTA under "Hard Mode"** (Addressing W91b, HLbn)

Reviewers inquired about the necessity of MGA given existing baselines like Nemotron. We addressed this by providing a granular breakdown of our reported results and conducting two supplementary experiments (Exp A.2 & C.2) to isolate specific variables. These comparisons reveal MGA's exceptional efficiency, particularly given the significant "Source Quality Deficit":

- **The "Source Gap":** MGA is derived from Fineweb-Edu, while Nemotron uses CC-HQ. Models trained on CC-HQ typically outperform Fineweb-Edu baselines by ~16 MMLU points (8B size with 1T training tokens).

- **Beating the Odds:** Despite this massive disadvantage in source data quality, MGA outperforms 5 out of 6 Nemotron subsets, including the WRAP-medium task baseline.

- **Parity in Aggregation:** In direct comparison (Exp C.2 vs Exp A), MGA achieves parity with Nemotron-Syn. This implies that as a methodology, MGA is likely more effective, as it is derived from a 'lower-quality' source domain.


### **Rebuttal Highlight II: Rigorous Methodological Validation** (Addressing eWsn)

We addressed the request for deeper validation of our framework design:

- **Human Alignment:** A human-in-the-loop audit confirmed our LLM-based quality filtering aligns with human judgment (>90% agreement).

- **Quantifying Diversity:**
    - We introduced Heterogeneity Score metrics, which prove our strategy effectively prevents mode collapse by control of prompts and judger feedback.
    - Additionally, we show that the one-pass-for-many design yields higher lexical and semantic diversity (Type-Token Ratio: 0.12 vs 0.08) compared to standard sampling.

- **Generalization to MoE:** We provided a validation to an internal MoE model family (6.8B/12.5B), confirming that MGA's benefits generalize beyond the dense models used in the main paper.

### **Conclusion**
This work offers a proven, scalable solution to data scarcity. By resolving the questions regarding baselines and diversity, we believe MGA stands as a robust and essential contribution to the community.

---

Once again, we extend our sincere gratitude to the reviewers for their detailed comments, which significantly improved the robustness of our work.
We also deeply appreciate the Area Chair’s time and effort in evaluating our work during this transition.

---

### Meta-Review · Area_Chair_QvcY · 2025-12-20

**Summary:**

This paper introduces a framework to augment data corpora by reformulating documents into multiple genre-audience pairs. Combining training sets with the proposed MGA yields effective models.

Reviews were mostly positive with asking for some clarifying experiments and ablations. The authors addressed many of these concerns reasonably. My overall reading of the paper and subsequent discussion is positive.

**Reviewer Concerns:**

This paper received 3 reviews:
- W91b: gave a weakly positive review with a key question asking about a fair comparison demonstrating that mixing with MGA outperforms using only Nemotron-Syn at the same budget.
- HLbn: gave a positive review with some concerns regarding baselines.
- eWsn: gave a weakly positive review with questions asking about the design choices of the prompt-based solution to synthetic data generation, and generalization to other architectures.

**Reviewer Scores:**

- W91b: I believe the reviewer would not change their score. The authors have performed the suggested experiment to show that mixing with MGA is not demonstrably worse than mixing with Nemotron-Syn at the same budget. Rather the two perform similarly. However, the authors explain that this type of direct comparison is confounded.
- HLbn: I believe the reviewer would not change their score. The authors provide a reasonable rebuttal to the reviewer’s questions, but do not demonstrate anything to suggest improving to the next level score.
- eWsn: I believe the reviewer would not decrease their score. The authors definitively address some of the reviewers questions, for example with empirics such as testing on other architectures.

---

### Decision · Program_Chairs · 2026-01-26

Accept (Poster)